# Quantum parameter estimation with many-body fermionic systems and application to the Hall effect

Olivier Giraud[1,2,3], Mark-Oliver Goerbig[4] and Daniel Braun[5]

**1** Université Paris-Saclay, CNRS, LPTMS, 91405 Orsay, France
**2** MajuLab, CNRS-UCA-SU-NUS-NTU International Joint Research Unit, Singapore
**3** Centre for Quantum Technologies, National University of Singapore, Singapore
**4** Université Paris-Saclay, CNRS, Laboratoire de Physique des Solides, 91405 Orsay, France
**5** Institut für theoretische Physik, Universität Tübingen, 72076 Tübingen, Germany

## Abstract

We calculate the quantum Fisher information for a generic many-body fermionic system in a pure state depending on a parameter. We discuss the situations where the parameter is imprinted in the basis states, in the state coefficients, or both. In the case where the parameter dependence of coefficients results from a Hamiltonian evolution, we derive a particularly simple expression for the quantum Fisher information. We apply our findings to the quantum Hall effect, and evaluate the quantum Fisher information associated with the optimal measurement of the magnetic field for a system in the ground state of the effective Hamiltonian. The occupation of electron states with high momentum enforced by the Pauli principle leads to a "super-Heisenberg" scaling of the sensitivity with a power law that depends on the geometry of the sensor.



## 1  Introduction

The Hall effect offers a precise and economic way of measuring magnetic fields with small, integrated sensors. Typical commercially available Hall sensors based on silicon have sensitivities of about $100\,\mathrm{nT/Hz}^{1/2}$ [1]. Graphene-based ones are projected to achieve sensitivities normalized to the width $w$ ($B_{\min}w$) of $4\,\mathrm{pT\cdot mm/}\sqrt{\mathrm{Hz}}$ at room temperature [2]. The quantum Hall effect, reached at very strong magnetic fields and low temperatures, has also become a cornerstone of metrology, allowing a measurement of the von Klitzing constant $R_K = h/e^2$ to 10 digits precision [3].

In the present work we do not investigate the precision with which one can access $R_K$, but assess the ultimate sensitivity of magnetic field sensors based on the quantum Hall effect. This ultimate sensitivity is only bound by quantum noise and thermal noise of the sensor, and should be attainable once all the technical noises, such as electrical noise in the amplifiers and wires, vibrations, fluctuating charges in the materials etc. have been removed. A powerful formalism for calculating this ultimate sensitivity is provided by the quantum Cramér-Rao bound (QCRB) [4–6], expressed in terms of the quantum Fisher information (QFI), which leads to an important ultimate benchmark of the sensitivity.

Motivated by the Hall effect application, we first investigate here more generally quantum parameter estimation with a system consisting of a large number of indistinguishable fermions (typically electrons). Such a system is most concisely described by fermionic quantum field theory, which we will briefly review in the following for setting up the notations used. We will consider quantum states written in a basis of many-particle states. These basis states are obtained by "creating" fermions in single-particle states, chosen here as eigenstates of some single-particle Hamiltonian. We will consider three different ways by which a parameter de-

pendence can be imprinted on such a state: via a parameter-dependent evolution Hamiltonian, via parameter-dependent fermionic many-particle basis states, or via a Bogoliubov transformation. Note that many other possibilities exist to imprint a parameter on a state, see [7]. In all three cases mentioned above, a parameter-independent initial state will be transformed into a parameter-dependent state via a parameter-dependent unitary transformation. Let us now discuss these three ways in turn.

*i*.) Time evolution generated by a Hamiltonian that depends on a parameter is the standard situation in most single-particle or single-mode applications of quantum metrology, where one typically considers a fixed (parameter-independent) basis and initial states, and time-dependent and parameter-dependent amplitudes for those. Both dependencies of the amplitudes arise from propagation with the Hamiltonian.

*ii*.) The parameter dependence may be imprinted in the many-particle basis states themselves. Indeed, as these are anti-symmetrized linear combinations of single-particle energy eigenstates, a change of the single-particle Hamiltonian can modify its eigenbasis, so that even without time evolution the state of the system can contain information about the value of the parameter. For a specific example, consider a single particle in a harmonic trap. Assume the particle is in the ground state of the oscillator, and the parameter we are interested in is the frequency of the trap. Through the oscillator length the ground-state wave function clearly depends on the frequency. Increasing the frequency squeezes the ground-state wave function in position space. Hence, even without time-evolution, one can measure, at least in principle, the frequency of the harmonic oscillator (see [8] for details). In quantum optics, the quantum fluctuations of the vacuum state (i.e. without any photons present) have indeed been measured directly [9], and it is clear that they depend on the frequency considered.

An underlying physical assumption of this reasoning is that the system is always in the actual parameter-dependent ground state (or any other state specified through a given number of excitations of a single-particle Hamiltonian or linear combinations thereof) when the parameter is changed. A change of the parameter must therefore happen adiabatically. However, according to the formalism of the quantum Cramér-Rao bound, only infinitesimal changes of the parameter need to be considered for determining the best sensitivity with which the parameter can be estimated, and hence adiabatic changes over infinitesimal changes of the parameter suffice to justify the model. We will therefore assume that the system indeed tracks the parameter-dependent many-particle states over infinitesimal changes of the parameter. This is a common assumption in the literature, see e.g. [10] and works citing it, where the QFI was calculated for a many-body ground state driven across a phase transition.

*iii*.) In the more general situation encountered in quantum field theory the number of particles need not be conserved, which creates an additional freedom for encoding parameters compared to single-particle quantum mechanics. Indeed, the most general linear transformations of the creation and annihilation operators that preserve their fermionic anti-commutation relations are Bogoliubov transformations. We will therefore consider Bogoliubov transformations as a third way of coding parameters in a state. In the most general situation, Bogoliubov transformations allow to mix excitations with creation of holes, which opens the way to a new kind of quantum parameter estimation not possible with single-particle basis change. We will first discuss this general case and derive very general expressions for the QFI. We will then consider the special case where the particle number is conserved, that is, when Bogoliubov transformations mix creation operators with creation operators only and annihilation operators with annihilation operators only. This corresponds to changing the single-particle basis states. This case will be relevant to application of our results to the quantum Hall effect.

Bogoliubov transformations for quantum parameter-estimation have been considered before in bosonic field theories [11]. Analytical results were obtained for the estimation of small parameters in terms of Bogoliubov coefficients for single-mode and two-mode Gaussian chan-

nels. The QFI for specific two-mode bosonic Gaussian states was also found in [12]. In [13] an exact expression for the QFI of an arbitrary two-mode bosonic Gaussian state was obtained. Carollo and co-workers calculated the symmetric logarithmic derivative of general Gaussian fermionic states [14]. In [15] a proper definition of entanglement in fermionic systems and its connection to the sensitivity of quantum metrology schemes based on them was investigated.

Here, we investigate quantum-parameter estimation for arbitrary pure states of indistinguishable fermions, and include all three ways of encoding a parameter described above. Performing a time evolution, a basis change or a Bogoliubov transformation amounts to applying a unitary operator to the initial quantum state. In Section 3 we calculate the QFI in the case where an initial parameter-independent state is subjected to a parameter-dependent unitary transformation. We then derive a chain rule for the QFI in the case of two successive unitary transformations, which allows us to identify the contribution from each of them as well as their mutual influence. In Section 4 we calculate the QFI for Bogoliubov transformations (whose formalism is reviewed in Section 2). Section 5 is dedicated to applying this formalism to the quantum Hall effect.

## 2 Fermionic quantum field theories and Bogoliubov transformations

The present section introduces some notation in Subsection 2.1 and recalls the Bogoliubov formalism in Subsections 2.2, 2.3 and 2.4. The reader familiar with this formalism can skip these latter subsections and go directly to Section 3.

### 2.1 Fermionic basis states

The most general pure state of indistinguishable fermions in $M$ single-particle modes can be written in the form

$$|\psi\rangle = \sum_{\boldsymbol{n}} \psi_{\boldsymbol{n}} |\boldsymbol{n}\rangle_c \,, \tag{1}$$

where the sum runs over all $M$-tuples $\boldsymbol{n} = (n_0, n_1, \ldots, n_{M-1})$ with $n_k \in \{0, 1\}$, and

$$|\boldsymbol{n}\rangle_c = (c_0^\dagger)^{n_0} (c_1^\dagger)^{n_1} \ldots (c_{M-1}^\dagger)^{n_{M-1}} |\mathrm{vac}\rangle_c \,, \tag{2}$$

which by convention we abbreviate to $\prod_{k=0}^{M-1} (c_k^\dagger)^{n_k} |\mathrm{vac}\rangle_c$, is the state of $n_k$ particles in mode $k$ for $k = 0, \ldots, M-1$. Here, the states $|\boldsymbol{n}\rangle_c$ are the $N$-particle states $|n_0\rangle_c \otimes |n_1\rangle_c \otimes \cdots \otimes |n_{M-1}\rangle_c$, or equivalently $|n_0, n_1, \ldots, n_{M-1}\rangle_c$, with $N = \sum_k n_k$, and $|n_k\rangle_c$ are eigenstates of a single-particle Hamiltonian, with $k = 0, 1, 2, \ldots$. The operator $c_k^\dagger$ creates a fermion in mode $k$ out of the vacuum $|\mathrm{vac}\rangle_c$. The vacuum state $|\mathrm{vac}\rangle_c$ of particles $c$ is defined as the state that satisfies $c_k |\mathrm{vac}\rangle_c = 0 \; \forall k = 0, \ldots, M-1$, i.e. it is a state that contains no particles of type $c$.

The set of all $2^M$ states $|\boldsymbol{n}\rangle_c$, with $\boldsymbol{n}$ running over all $M$-tuples of 0 and 1, forms a basis of Fock space, and $\psi_{\boldsymbol{n}}$ in (1) are the (complex) coefficients of $|\psi\rangle$ in that basis. We will consider $|\psi\rangle$ to depend on a parameter $\omega$ that we want to estimate. In the most general situation the parameter dependence can arise both from the $\psi_{\boldsymbol{n}}$ and from the basis states $|\boldsymbol{n}\rangle_c$. Note that for energies much smaller than the rest masses of the fermions, superpositions containing a different number of particles are forbidden by the particle-number superselection rule. States of the form (1) are nevertheless considered for example in BCS theory of superconductivity [16], where particle number conservation is enforced only on average (and to a very good precision, for a large number of particles). Of course, by an appropriate choice of the $\psi_{\boldsymbol{n}}$, one can restrict $|\psi\rangle$ to a state with a fixed number of particles.

## 2.2 Bogoliubov transformations

We consider the situation where the $c_k$ arise from a parameter-dependent Bogoliubov transformation from parameter-independent creation and annihilation operators $a_k^\dagger$ and $a_k$. Bogoliubov transformations are the most general linear transformations that preserve canonical anticommutation relations. They take the general form

$$c_i^\dagger = a_j^\dagger U_{ji} + a_j V_{ji},$$
$$c_i = a_j U_{ji}^* + a_j^\dagger V_{ji}^* \tag{3}$$

(with Einstein summation convention), where $U_{ji}, V_{ji}$ are parameter-dependent complex numbers. The preservation of the anticommutation relations $\{c_i^\dagger, c_j\} = \{a_i^\dagger, a_j\} = \delta_{ij}$ implies the condition $U^\dagger U + V^\dagger V = \mathrm{Id}_M$, while $\{c_i, c_j\} = \{a_i, a_j\} = 0$ gives $U^t V + V^t U = 0$, where $V^t$ denotes the transpose of $V$, and $\mathrm{Id}_M$ the $M \times M$-dimensional identity matrix. When arranged as a matrix $W$ with

$$W = \begin{bmatrix} U & V^* \\ V & U^* \end{bmatrix}, \tag{4}$$

the two above conditions on $U$ and $V$ can be equivalently expressed as $W^\dagger W = \mathrm{Id}_{2M}$, so that $W$ is unitary. Following [17] we introduce the compact vector notation $\alpha = (a^\dagger, a) \equiv (a_0^\dagger, \ldots a_{M-1}^\dagger, a_0, \ldots a_{M-1})$, and correspondingly $\gamma = (c^\dagger, c) \equiv (c_0^\dagger, \ldots c_{M-1}^\dagger, c_0, \ldots c_{M-1})$. We shall denote by $\alpha^\dagger$ the column vector $(a_0, \ldots a_{M-1}, a_0^\dagger, \ldots a_{M-1}^\dagger)^T$. The Bogoliubov transformation (3) can then be written simply as $\gamma = \alpha W$.

Let $S$ be the $2M \times 2M$ matrix defined from $W$ by the relation

$$W = \exp(iS\Xi), \tag{5}$$

with

$$\Xi = \begin{bmatrix} 0 & \mathrm{Id}_M \\ \mathrm{Id}_M & 0 \end{bmatrix}. \tag{6}$$

Because of (5) and the definition (4), the matrix $S$ has the block form

$$S = \begin{bmatrix} S^{(2)} & S^{(1)} \\ -S^{(1)*} & -S^{(2)*} \end{bmatrix}, \tag{7}$$

where $S^{(1)}$ and $S^{(2)}$ are (in general complex) $M \times M$ matrices, with $S^{(1)} = S^{(1)\dagger}$ Hermitian and $S^{(2)} = -S^{(2)t}$ antisymmetric. The matrix $S$ is not Hermitian in general, but it satisfies $S^t = -S$, and hence $S^\dagger = -S^*$. We define the operators

$$\hat{S}(\alpha) = \frac{1}{2} \alpha S \alpha^t, \tag{8}$$

$$\hat{T}(\alpha) = \exp(i\hat{S}(\alpha)). \tag{9}$$

Since $\alpha^t = \Xi \alpha^\dagger$ (where the $^\dagger$ conjugates the annihilation and creation operators and transforms the row vector into a column vector), $\hat{S}(\alpha)$ can be written in the alternative form $\hat{S}(\alpha) = \frac{1}{2} \alpha S \Xi \alpha^\dagger$. The operator $\hat{T}(\alpha)$ satisfies the identity

$$\hat{T}(\alpha) \alpha \hat{T}(\alpha)^\dagger = \alpha W = \gamma. \tag{10}$$

## 2.3 Relation between bases

To the vacuum state $|vac\rangle_c$ for particles of type $c$ corresponds a vacuum state $|vac\rangle_a$ for particles of type $a$. It is defined by $a_k |vac\rangle_a = 0 \ \forall k = 0, \ldots, M-1$. In general, the two vacua are different, $|vac\rangle_c \neq |vac\rangle_a$, as is obvious from the fact that whenever $V_{ji} \neq 0$ in Eq. (3), the operator $c_i$ creates a particle of type $a$. The two vacua are related via

$$|vac\rangle_c = \hat{T}(\alpha) |vac\rangle_a \, , \tag{11}$$

as can be readily seen by noting that Eqs. (10) and (11) imply

$$c_k |vac\rangle_c = \hat{T}(\alpha) a_k \hat{T}^\dagger(\alpha) \hat{T}(\alpha) |vac\rangle_a = \hat{T}(\alpha) a_k |vac\rangle_a = 0 \tag{12}$$

(see e.g. [17]). Only in the case where $V = 0$ (i.e. the Bogoliubov transformation does not mix creation operators with annihilation operators) does one have, up to possibly a phase, $|vac\rangle_c = |vac\rangle_a$. Equation (11) generalizes to an arbitrary state: one has for a Fock state

$$
\begin{aligned}
|\boldsymbol{n}\rangle_c &= \prod_{k=0}^{M-1} (c_k^\dagger)^{n_k} |vac\rangle_c \\
&= \prod_{k=0}^{M-1} \left( \hat{T}(\alpha)(a_k^\dagger)^{n_k} \hat{T}(\alpha)^\dagger \right) \hat{T}(\alpha) |vac\rangle_a \\
&= \hat{T}(\alpha) \prod_{k=0}^{M-1} (a_k^\dagger)^{n_k} |vac\rangle_a \\
&= \hat{T}(\alpha) |\boldsymbol{n}\rangle_a \, ,
\end{aligned}
\tag{13}
$$

and by linearity for an arbitrary pure state

$$|\psi\rangle = \sum_{\boldsymbol{n}} \psi_{\boldsymbol{n}} |\boldsymbol{n}\rangle_c = \hat{T}(\alpha) \sum_{\boldsymbol{n}} \psi_{\boldsymbol{n}} |\boldsymbol{n}\rangle_a \, . \tag{14}$$

## 2.4 One-particle overlaps

We now calculate the overlap between one-particle states in terms of the Bogoliubov parameters. Let $R$ be the $M \times M$ matrix defined as

$$R_{kl} = {}_a \langle k | l \rangle_c \, , \tag{15}$$

where $|k\rangle_c$ is the state with one particle in mode $k$, i.e. the state $|\boldsymbol{n}\rangle_c$ with $n_i = \delta_{ik}, 0 \leq i \leq M-1$. Using the expression of $\gamma_i$ given by Eq. (10), we have for $i, j \in \{1, \ldots, 2M\}$

$$
{}_a \langle vac| \, \alpha_j \gamma_i \, |vac\rangle_c = {}_a \langle vac| \, \alpha_j \alpha_k \, |vac\rangle_c \, W_{ki} \tag{16}
$$

(again with implicit summation). Since by definition $a_j |vac\rangle_a = |0\rangle$ and $c_i |vac\rangle_c = |0\rangle$, the left-hand side of (16) has the block structure

$$
\begin{bmatrix} 0 & 0 \\ R & 0 \end{bmatrix} . \tag{17}
$$

On the right-hand side of (16) the term ${}_a \langle vac| \, \alpha_j \alpha_k \, |vac\rangle_c$ has the block structure

$$
\begin{bmatrix} 0 & 0 \\ P & Q \end{bmatrix} . \tag{18}
$$

Using the block structure (4) of $W$, Eq. (16) readily gives

$$PU + QV = R, \tag{19}$$

$$PV^* + QU^* = 0. \tag{20}$$

Matrices $P$ and $Q$ can be calculated by using the following canonical decomposition for the operators $\hat{T}(\alpha)$ [17]:

$$\hat{T}(\alpha) = |U^\dagger|^{1/2} e^{\hat{Z}} e^{\hat{Y}} e^{\hat{X}}, \tag{21}$$

where

$$\hat{X} = \frac{1}{2}\sum_{i,j} X_{ij} a_i a_j, \quad X = U^{*-1} V, \tag{22}$$

$$\hat{Y} = \frac{1}{2}\sum_{i,j} Y_{ij} a_i^\dagger a_j, \quad e^{-Y} = U^\dagger, \tag{23}$$

$$\hat{Z} = \frac{1}{2}\sum_{i,j} Z_{ij} a_i^\dagger a_j^\dagger, \quad Z = V^* U^{*-1}, \tag{24}$$

and $|.|$ denotes the determinant (recall that in general $U$ is not a unitary matrix). While operators $\hat{X}$ and $\hat{Y}$ contain annihilation operators, $\hat{Z}$ only contains creation operators. Thus $|\text{vac}\rangle_c = \hat{T}(\alpha)|\text{vac}\rangle_a = |U^\dagger|^{1/2} e^{\hat{Z}}|\text{vac}\rangle_a$ and the overlap between vacua reads

$$_a\langle\text{vac}|\text{vac}\rangle_c = |U^\dagger|^{1/2}. \tag{25}$$

Matrix $P$ is readily obtained as

$$P_{kj} = {}_a\langle\text{vac}| a_k a_j^\dagger |\text{vac}\rangle_c = |U^\dagger|^{1/2}\delta_{kj}. \tag{26}$$

Using the identity [18]

$$\left[a_k, e^{\hat{Z}}\right] = \sum_l Z_{kl} a_l^\dagger e^{\hat{Z}} \tag{27}$$

(which can be shown by induction), we get

$$\begin{aligned}
_a\langle\text{vac}| a_n a_k e^{\hat{Z}}|\text{vac}\rangle_a &= {}_a\langle\text{vac}| a_n \left[a_k, e^{\hat{Z}}\right]|\text{vac}\rangle_a \\
&= \sum_l {}_a\langle\text{vac}| a_n a_l^\dagger e^{\hat{Z}}|\text{vac}\rangle_a Z_{kl} \\
&= {}_a\langle\text{vac}| e^{\hat{Z}}|\text{vac}\rangle_a Z_{kn} \\
&= -Z_{nk}
\end{aligned} \tag{28}$$

(the relation $U^t V + V^t U = 0$ implies $Z^t = -Z$). Therefore,

$$Q_{nk} = {}_a\langle\text{vac}| a_n a_k |\text{vac}\rangle_c = -Z_{nk}|U^\dagger|^{1/2}. \tag{29}$$

When we replace $P$ and $Q$ by their above expression in Eq. (20), we get $V^* - ZU^* = 0$, which is in fact a direct consequence of the definition of $Z$. Doing the same in Eq. (19) we get $|U^\dagger|^{1/2}(U - ZV) = R$. Using (24), this gives

$$R = |U^\dagger|^{1/2}\left(U - V^* U^{*-1} V\right). \tag{30}$$

We recognize the Schur complement of the block $U^*$ in matrix $W$, which appears in the expression for the inverse of the block matrix $W$. Since $W$ is unitary, Eq. (30) reduces to

$$R = |U^\dagger|^{1/2} U^{\dagger-1}. \tag{31}$$

If the Bogoliubov transformation is such that $V = 0$, then the relation $U^\dagger U + V^\dagger V = \text{Id}_M$ implies that $U$ is unitary. From Eq. (31) we then have $U = R$, so that for this particular Bogoliubov transformation $U$ is simply the matrix of one-particle overlaps. In other words, the Bogoliubov transformation between two single-particle bases can be obtained by taking $U = R$ and $V = 0$. This is the situation we encounter in section 4.1.4 below.

# 3 Quantum Cramér-Rao bound and quantum Fisher information in fermionic quantum field theories

## 3.1 Quantum Cramér-Rao bound

Let $|\psi_\omega\rangle$ be a quantum state which depends on a parameter $\omega$. More generally, consider a density matrix $\rho(\omega)$ that describes a parameter-dependent mixed state. One would like to know how precisely one can estimate $\omega$ based on the measurement of some observables. This will depend in general, of course, on a lot of things, starting with the measurements chosen, the precision of the measurement devices used, the noisiness of the environment, the number of measurements, the statistics of the data obtained, and how the data are analyzed. However, with the quantum Cramér-Rao bound (QCRB) [4–6], a very powerful theoretical tool is available that allows one to calculate the smallest possible uncertainty of any unbiased estimate of $\omega$, no matter what positive-operator-valued measure (POVM) measurements are performed, and what estimator functions are used to analyze the data, as long as they are unbiased estimator functions based on the measurement results alone. Suppose we want to estimate a parameter $\omega$ by measuring $M_e$ times a quantity $X$ (e.g. a POVM) whose statistics of outcomes $P(X = x|\omega)$ depends on $\omega$. An estimator $\hat{\omega}(x_1, \ldots, x_{M_e})$ is any function that maps the $M_e$ measurement results $x_1, \ldots, x_{M_e}$ to an estimate of the parameter $\omega$. It is called unbiased if the average of $\hat{\omega}$ for the probability distribution $P(X = x|\omega)$ is $\langle\hat{\omega}\rangle = \omega$ locally. With such an estimate at hand, measurement of $X$ allows one to access $\omega$. However, since the measurement results fluctuate in general due to the quantum nature of the state, so does the estimator. Its smallest possible variance gives the optimal sensitivity with which one can estimate $\omega$ by measuring $X$. The QCRB provides a lower bound for the variance of $\hat{\omega}$. It reads

$$\mathrm{var}(\hat{\omega}) \geq \frac{1}{M_e} \frac{1}{I(\rho(\omega), \omega)}, \tag{32}$$

where $I$ is the quantum Fisher information, given by

$$I(\rho(\omega), \omega) = \mathrm{tr}\big(\rho(\omega)L^2(\omega)\big), \tag{33}$$

and the symmetric logarithmic derivative operator $L(\omega)$ is a linear operator defined through

$$\partial_\omega \rho(\omega) = \frac{1}{2}(L(\omega)\rho(\omega) + \rho(\omega)L(\omega)). \tag{34}$$

The bound can be saturated in the limit of $M_e \to \infty$. The QFI can generically [19, 20] be interpreted geometrically through the Bures distance between two states $\rho(\omega)$ and $\rho(\omega + \delta\omega)$ that differ infinitesimally in the parameter. This gives an appealing physical interpretation to the QCRB: the ultimate sensitivity with which a parameter coded in a quantum state can be estimated is all the more large as the state depends strongly on the parameter.

## 3.2 General expressions for the quantum Fisher information

The QFI for systems with infinite-dimensional Hilbert space is in general difficult to calculate, as it typically requires the diagonalization of the density matrix in order to determine the logarithmic derivative or the calculation of the Bures distance. However, if the state is given already in diagonalized form, closed expressions for the QFI can be found. The simplest case in this category is that of a pure state $|\psi_\omega\rangle$. Its QFI can be shown to be [21]

$$I(|\psi_\omega\rangle, \omega) = 4\big(\langle\dot{\psi}_\omega|\dot{\psi}_\omega\rangle + \langle\dot{\psi}_\omega|\psi_\omega\rangle^2\big), \tag{35}$$

where $|\dot{\psi}_\omega\rangle \equiv \partial_\omega |\psi_\omega\rangle$ (see Eq. (26) in [22]). Note that in the whole paper dots denote derivatives with respect to the parameter $\omega$. For a mixed state $\rho(\omega)$ given in its eigenbasis, $\rho(\omega) = \sum_r p_r(\omega) |\psi_\omega^{(r)}\rangle\langle\psi_\omega^{(r)}|$, where the $|\psi_\omega^{(r)}\rangle$ form an orthonormal basis, one has

$$I(\rho(\omega),\omega) = \sum_r \frac{(\partial_\omega p_r)^2}{p_r} + 2\sum_{n,m} \frac{(p_n - p_m)^2}{p_n + p_m} \left|\langle\psi_\omega^{(n)}|\dot{\psi}_\omega^{(m)}\rangle\right|^2 , \tag{36}$$

where the sums are over all terms with non-vanishing denominators.

The form (35) can be equivalently expressed as

$$I(|\psi_\omega\rangle,\omega) = 4\langle\dot{\psi}_\omega|(\mathbb{1} - |\psi_\omega\rangle\langle\psi_\omega|)|\dot{\psi}_\omega\rangle. \tag{37}$$

Under that form, the QFI can be directly related to the Fubini-Study metric. More generally, the QFI has a simple geometric interpretation: it is related to the Bures distance between two infinitesimally close states [23] via the identity [24]

$$I(\rho(\omega),\omega) = 4\lim_{\delta\omega\to 0} \frac{d_B(\rho(\omega),\rho(\omega+\delta\omega))^2}{\delta\omega^2}, \tag{38}$$

with

$$d_B(\rho,\rho') = \left(2 - 2\operatorname{tr}\sqrt{\rho^{\frac{1}{2}}\rho'\rho^{\frac{1}{2}}}\right)^{\frac{1}{2}}. \tag{39}$$

## 3.3 QFI for a unitary transformation

### 3.3.1 General pure state

The most general pure states of a system described within quantum field theory are of the form (1). Both the basis states $|n\rangle_c$ and the amplitudes $\psi_n(\omega)$ can depend on the parameter $\omega$, so that we have to deal with states of the form

$$|\psi_\omega\rangle = \sum_n \psi_n(\omega)|n\rangle_\omega. \tag{40}$$

The reason for this is that the basis states $|n\rangle_\omega$ are constructed as antisymmetrized linear combinations of single-particle eigenstates that can depend on the parameter through the single-particle Hamiltonian. For example, in the case of the quantum Hall effect that we will consider in detail in section 5, the single-particle energy eigenstates correspond to Landau levels that depend on the magnetic field (or equivalently the cyclotron frequency $\omega$), i.e. they are energy eigenstates of an harmonic oscillator with frequency $\omega$, leading e.g. in position representation to wavefunctions with a typical length scale given by the frequency-dependent oscillator length.

As mentioned in the Introduction, a parameter can be imprinted on a quantum state via an evolution Hamiltonian, via many-particle basis states even in the absence of a time evolution, or via a Bogoliubov transformation. A parameter-independent initial state is transformed into a parameter-dependent state via a parameter-dependent unitary transformation. In addition, the propagation of a superposition of eigenstates leads to parameter-dependent phases of the amplitudes. As we will show, the form (40) can be obtained from a parameter-independent state by means of two consecutive unitary operators. We first consider the case where a single unitary operator is applied. Of course, one could always combine these two unitaries into a single one, but for some applications the decomposition into two unitaries is natural, as will be illustrated in Sec. 4.2 below.

### 3.3.2 A single unitary

Suppose the unitary transformation is of the form $\hat{T}_\omega = \exp(i\hat{S}_\omega)$, with $\hat{S}_\omega$ Hermitian, acting on some parameter-independent reference state $\left|\psi_{\omega_0}\right\rangle$, so that the state is of the form $|\psi_\omega\rangle = \hat{T}_\omega \left|\psi_{\omega_0}\right\rangle$. This situation arises for example by a time evolution driven by a Hamiltonian $\hat{H}_\omega$ that depends on the parameter $\omega$ (in which case $\hat{S}_\omega = -\hat{H}_\omega t$ is also proportional to time $t$). A simple calculation shows that the QFI (35) can be rewritten as

$$I\left(|\psi_\omega\rangle, \omega\right) = 4\operatorname{var}\left(\mathcal{H}, \left|\psi_{\omega_0}\right\rangle\right), \quad \text{with} \quad \mathcal{H} = -i\hat{T}_\omega^\dagger \dot{\hat{T}}_\omega, \tag{41}$$

where the operator $\mathcal{H}$ is Hermitian and

$$\operatorname{var}(\mathcal{H}, |\psi\rangle) = \langle\psi| \mathcal{H}^2 |\psi\rangle - \langle\psi| \mathcal{H} |\psi\rangle^2. \tag{42}$$

### 3.3.3 Two unitaries and chain rule for the QFI

Let us now consider the case where the parameter is encoded in $|\psi_\omega\rangle$ by means of two consecutive unitaries depending on $\omega$. Our aim is thus to calculate the QFI of a state of the form

$$|\psi_\omega\rangle = \hat{U}_\omega \hat{T}_\omega \left|\psi_{\omega_0}\right\rangle. \tag{43}$$

In the same way as $\mathcal{H}$ in Eq. (41), we define $\mathcal{U} = -i\hat{U}_\omega^\dagger \dot{\hat{U}}_\omega$. From unitarity of $\hat{U}_\omega$ and $\hat{T}_\omega$ we have

$$\mathcal{H} = -i\hat{T}_\omega^\dagger \dot{\hat{T}}_\omega = i\dot{\hat{T}}_\omega^\dagger \hat{T}_\omega, \qquad \mathcal{H}^2 = \dot{\hat{T}}_\omega^\dagger \dot{\hat{T}}_\omega, \tag{44}$$

$$\mathcal{U} = -i\hat{U}_\omega^\dagger \dot{\hat{U}}_\omega = i\dot{\hat{U}}_\omega^\dagger \hat{U}_\omega, \qquad \mathcal{U}^2 = \dot{\hat{U}}_\omega^\dagger \dot{\hat{U}}_\omega, \tag{45}$$

with $\mathcal{H}$ and $\mathcal{U}$ Hermitian. We introduce the state

$$|\phi_\omega\rangle = \hat{T}_\omega \left|\psi_{\omega_0}\right\rangle, \tag{46}$$

so that

$$|\psi_\omega\rangle = \hat{U}_\omega |\phi_\omega\rangle, \tag{47}$$

$$|\dot{\psi}_\omega\rangle = \dot{\hat{U}}_\omega |\phi_\omega\rangle + \hat{U}_\omega |\dot{\phi}_\omega\rangle, \tag{48}$$

$$|\dot{\phi}_\omega\rangle = \dot{\hat{T}}_\omega \left|\psi_{\omega_0}\right\rangle. \tag{49}$$

This yields the identities

$$\langle\dot{\phi}_\omega|\dot{\phi}_\omega\rangle = \langle\psi_{\omega_0}|\mathcal{H}^2|\psi_{\omega_0}\rangle, \tag{50}$$

$$\langle\dot{\phi}_\omega|\phi_\omega\rangle = -i\langle\psi_{\omega_0}|\mathcal{H}|\psi_{\omega_0}\rangle. \tag{51}$$

From Eq. (35) we then obtain

$$\frac{1}{4}I(|\psi_\omega\rangle, \omega) = \operatorname{var}(\mathcal{H}, \left|\psi_{\omega_0}\right\rangle) + \operatorname{var}(\mathcal{U}, |\phi_\omega\rangle) \tag{52}$$

$$- 2\operatorname{Im}\langle\dot{\phi}_\omega|\mathcal{U}|\phi_\omega\rangle - 2\langle\dot{\phi}_\omega|\mathcal{U}|\phi_\omega\rangle \langle\psi_{\omega_0}|\mathcal{H}|\psi_{\omega_0}\rangle.$$

Equation (52) provides a chain rule for the QFI associated with two unitary operators. If $\hat{U}_\omega$ or $\hat{T}_\omega$ is the identity operator, one gets back the expression (41) for a single operator. When two unitaries are present, the variances sum up, but in addition there is a cross term that comes from the variation of both $\hat{U}_\omega$ and $\hat{T}_\omega$ with the parameter. Note that recently, a chain rule for the QFI was derived in a different context, namely in a case where quantum evolution is followed by a POVM that depends itself on the parameter [25].

# 4 Some specific cases

## 4.1 QFI for Bogoliubov transformations

We now consider the situation where the parameter $\omega$ is encoded in $|\psi_\omega\rangle$ by means of a single unitary transformation $\hat{T}_\omega(\alpha)$ associated with a Bogoliubov transformation. The operator $\hat{T}_\omega(\alpha)$ is defined by Eqs. (4)–(9), with matrices $W$ and $S$ depending on a parameter $\omega$. In the language of section 3.3, particles of type $c$ correspond to parameter value $\omega$ and particles of type $a$ to parameter value $\omega_0$.

This situation is a special case of section 3.3 where the operator $\hat{S}_\omega$ is quadratic in creation and annihilation operators. The QFI is thus directly given by Eq. (41), where the Hermitian operator $\mathcal{H}$ is $\mathcal{H} = -i\hat{T}_\omega(\alpha)^\dagger \dot{\hat{T}}_\omega(\alpha)$. Our aim is to reexpress the QFI in terms of the matrices $U$ and $V$ defining the Bogoliubov transformation.

### 4.1.1 General case

Using Eq. (A.4) giving the derivative of an integral, we first rewrite $\mathcal{H}$ as [24, 26]

$$\mathcal{H} = \int_0^1 ds\, e^{-is\hat{S}_\omega(\alpha)} \frac{d\hat{S}_\omega(\alpha)}{d\omega} e^{is\hat{S}_\omega(\alpha)} \tag{53}$$

$$= \frac{1}{2}\dot{S}_{ij} \int_0^1 ds\, e^{-is\hat{S}_\omega(\alpha)}\alpha_i e^{is\hat{S}_\omega(\alpha)} e^{-is\hat{S}_\omega(\alpha)}\alpha_j e^{is\hat{S}_\omega(\alpha)}$$

(again with implicit summation over repeated indices). The term $e^{-is\hat{S}_\omega(\alpha)}\alpha_i e^{is\hat{S}_\omega(\alpha)}$ can be rewritten as $\hat{T}_\omega(\alpha)^{-s}\alpha_i \hat{T}_\omega(\alpha)^s = (\alpha e^{-isS\Xi})_i$, yielding

$$\mathcal{H} = \frac{1}{2}\int_0^1 ds\,(\alpha e^{-isS\Xi})_i \dot{S}_{ij}(\alpha e^{-isS\Xi})_j \tag{54}$$

$$= \frac{1}{2}\alpha_l \int_0^1 ds\,(e^{-isS\Xi})_{li}\dot{S}_{ij}(e^{-isS\Xi})_{kj}\alpha_k \tag{55}$$

$$= \frac{1}{2}\alpha_l \int_0^1 ds\,(e^{-isS\Xi})_{li}\dot{S}_{ij}(e^{is\Xi S})_{jk}\alpha_k\,, \tag{56}$$

where between (55) and (56) we have used $(S\Xi)^t = -\Xi S$ due to antisymmetry of $S$. The operator $\mathcal{H}$ can thus be expressed as a quadratic form in the $a_i, a_i^\dagger$ as

$$\mathcal{H} = \frac{1}{2}\alpha\tilde{\Omega}\alpha^t\,, \tag{57}$$

with

$$\tilde{\Omega} = \int_0^1 ds\, e^{-isS\Xi}\dot{S}e^{is\Xi S}\,. \tag{58}$$

In Appendix A we give an alternative proof of (58) based on the commutation relations of $\hat{S}$ and $\dot{\hat{S}}$. The above equation gives the most general expression for the operator whose variance gives the QFI. The remaining integral in Eq. (58) makes it uneasy to use. In order to make some progress we now consider a natural additional hypothesis.

### 4.1.2  Case $[S, \Xi] = 0$

The general result (57)–(58) can be further simplified if we make the additional assumption that $S$ and $\Xi$ commute: The block structure (7) implies that $[S, \Xi] = 0$ if and only if $S = -S^*$, that is, $iS$ is a real matrix. In such a case, using (A.4), Eq. (58) gives

$$\tilde{\Omega}\Xi = \int_0^1 ds \, e^{-isS\Xi} \dot{S}\Xi e^{isS\Xi} = -iW^\dagger \dot{W}, \tag{59}$$

so that $\mathcal{H}$ becomes

$$\mathcal{H} = \frac{1}{2}\alpha\Omega\alpha^\dagger, \quad \text{with} \quad \Omega = -iW^\dagger \dot{W} \tag{60}$$

(we used the identity $\Xi\alpha^t = \alpha^\dagger$ mentioned below Eq. (9)). Thus, in such a case where $iS$ is real, the matrix $W$ that defines the Bogoliubov transformation, together with its derivative with respect to the parameter $\omega$, provide an expression for $\mathcal{H}$ as a quadratic form of the operators $a_i^\dagger, a_i$.

Below, we will be interested in the calculation of the QFI as a function of $\omega$ in the vicinity of a fixed parameter $\omega_0$. We will therefore evaluate all quantities in the limit $\omega \to \omega_0$. In this limit, the Bogoliubov transformation goes to the identity, so that we have $U \to \mathrm{Id}_M$ and $V \to 0$. The matrix $\dot{W}$ then involves derivatives of $U$ and $V$ with respect to $\omega$ taken at $\omega \to \omega_0$, that we will denote $\dot{U}$ and $\dot{V}$. Taking the derivative of the relations $U^\dagger U + V^\dagger V = \mathrm{Id}_M$ and $U^t V + V^t U = 0_M$ with respect to $\omega$ and then the limit $\omega \to \omega_0$ we get $\dot{U} + \dot{U}^t = 0$ and $\dot{V} + \dot{V}^t = 0$; this can be shown by using the fact that since $iS$ is real then $W = \exp(iS\Xi)$ is, too (with $\Xi$ defined by Eq. (6)), and therefore also $U$ and $V$. With this antisymmetry of $\dot{U}$ and $\dot{V}$ together with the fermionic anticommutation relations, Eq. (60) becomes

$$\mathcal{H} = -i\sum_{k<l}\left(\dot{U}_{kl}a_k^\dagger a_l - \dot{U}_{kl}a_l^\dagger a_k + \dot{V}_{kl}a_k^\dagger a_l^\dagger + \dot{V}_{kl}a_k a_l\right). \tag{61}$$

### 4.1.3  Case $U = R$ real and $V = 0$

From now on we will specialize to the case where the Bogoliubov transformation does not mix creation and annihilation operators, i.e. $V = 0$, and the unitary transformation $U$ is orthogonal. This case is of great relevance, since it is precisely the framework in which we will derive expressions in Section 5. Indeed, in the context of the quantum Hall effect the Bogoliubov transformation is given by the matrix $R$ of overlaps (C.1), whose entries are real. The fact that these overlaps are real is a consequence of the structure of the Hall wavefunctions in the Landau gauge, given by Eq. (101) below: the complex phase is a plane wave that does not depend on the parameter $\omega$, yielding a (real) delta function in the overlap.

In this case, $W$ is real and one can always choose a matrix $S$ in (5) such that $iS\Xi$ is real. Hence $iS$ is real, and as a consequence $[S, \Xi] = 0$. Equation (61) can thus be used, and it gives

$$\mathcal{H} = -i\sum_{k<l}\dot{R}_{kl}(a_k^\dagger a_l - a_l^\dagger a_k). \tag{62}$$

### 4.1.4  QFI for a basis state

Let us consider the case where $|\psi_\omega\rangle$ is the parameter-dependent basis state $|\boldsymbol{n}\rangle_\omega = \hat{T}_\omega |\boldsymbol{n}\rangle_{\omega_0}$. Again we associate mode $c$ with $\omega$ and mode $a$ with $\omega_0$. According to (41), the QFI is given by the variance $I(|\boldsymbol{n}\rangle_\omega, \omega) = 4\mathrm{var}(\mathcal{H}, |\boldsymbol{n}\rangle_{\omega_0})$.

In the remainder of the paper we will only address the case where $[S, \Xi] = 0$, in which case $\mathcal{H}$ is given by the expression (61). It only requires to calculate $\mathcal{H} |\mathbf{n}\rangle_a$. We have

$$a_k |\mathbf{n}\rangle_a = \delta_{n_k,1} (-1)^{\sum_{j=0}^{k-1} n_j} |\overline{\mathbf{n}}^k\rangle_a , \tag{63}$$

$$a_k^\dagger |\mathbf{n}\rangle_a = \delta_{n_k,0} (-1)^{\sum_{j=0}^{k-1} n_j} |\overline{\mathbf{n}}^k\rangle_a , \tag{64}$$

where $|\overline{\mathbf{n}}^k\rangle_a$ is the state $|\mathbf{n}\rangle_a$ with $n_k$ replaced by $1 - n_k$ (i.e. the $k$th "bit" in the binary string $\mathbf{n}$ is flipped). This leads (for $k < l$) to

$$a_k^\dagger a_l |\mathbf{n}\rangle_a = \delta_{n_k,0} \delta_{n_l,1} (-1)^{\sum_{j=k}^{l-1} n_j} |\overline{\mathbf{n}}^{k,l}\rangle_a , \tag{65}$$

$$a_l^\dagger a_k |\mathbf{n}\rangle_a = -\delta_{n_k,1} \delta_{n_l,0} (-1)^{\sum_{j=k}^{l-1} n_j} |\overline{\mathbf{n}}^{k,l}\rangle_a , \tag{66}$$

where in $|\overline{\mathbf{n}}^{k,l}\rangle_a$ bits $k$ and $l$ are flipped. Similarly we have (still for $k < l$)

$$a_k^\dagger a_l^\dagger |\mathbf{n}\rangle_a = \delta_{n_k,0} \delta_{n_l,0} (-1)^{\sum_{j=k}^{l-1} n_j} |\overline{\mathbf{n}}^{k,l}\rangle_a , \tag{67}$$

$$a_k a_l |\mathbf{n}\rangle_a = \delta_{n_k,1} \delta_{n_l,1} (-1)^{\sum_{j=k}^{l-1} n_j} |\overline{\mathbf{n}}^{k,l}\rangle_a . \tag{68}$$

Inserting these expressions into Eq. (61) leads to

$$\mathcal{H} |\mathbf{n}\rangle_a = -i \sum_{k<l} \left( \dot{U}_{kl} \delta_{n_k,0} \delta_{n_l,1} + \dot{U}_{kl} \delta_{n_k,1} \delta_{n_l,0} + \dot{V}_{kl} \delta_{n_k,0} \delta_{n_l,0} + \dot{V}_{kl} \delta_{n_k,1} \delta_{n_l,1} \right) (-1)^{\sum_{j=k}^{l-1} n_j} |\overline{\mathbf{n}}^{k,l}\rangle_a , \tag{69}$$

which can be shortened to

$$\mathcal{H} |\mathbf{n}\rangle_a = -i \sum_{k<l} (-1)^{\sum_{j=k}^{l-1} n_j} D_{kl}^{(|n_k - n_l|)} |\overline{\mathbf{n}}^{k,l}\rangle_a , \tag{70}$$

with $D^{(0)} = \dot{V}$ and $D^{(1)} = \dot{U}$. Since each flipped state $|\overline{\mathbf{n}}^{k,l}\rangle_a$ is orthogonal to $|\mathbf{n}\rangle_a$, we have $_a\langle \mathbf{n}| \mathcal{H} |\mathbf{n}\rangle_a = 0$. The quadratic term in (42) is given by the square of the 2-norm of $\mathcal{H} |\mathbf{n}\rangle_a$. Since all terms in the sum (70) are orthogonal to each other, the QFI finally reads

$$I(|\mathbf{n}\rangle_\omega , \omega) = 4 \sum_{k<l} |D_{kl}^{(|n_k - n_l|)}|^2 . \tag{71}$$

In the case where $U = R$ is real and $V = 0$, $\mathcal{H}$ is given by Eq. (62). Only $D^{(1)} = \dot{R}$ contributes, so that Eq. (71) reduces to

$$I(|\mathbf{n}\rangle_\omega , \omega) = 4 \sum_{\substack{k<l \\ |n_k - n_l| = 1}} |\dot{R}_{kl}|^2 . \tag{72}$$

This is the expression which we shall use in Section 5 in the context of the quantum Hall effect.

It is interesting to analyze Eq. (72) in the context of a finite-dimensional Hilbert space. The sum in (72) is a sum over all pairs of occupied and unoccupied modes. For a finite-dimensional Hilbert space of single-particle states where each state is occupied (e.g. an insulating band in a solid), this sum vanishes. Indeed, as the corresponding Fock space is one-dimensional, all parameter dependence through unitary transformations amongst the annihilators trivially reduces to a phase, which cancels in the density matrix. Hence, the state is independent of the parameter under such unitaries, as can be checked explicitly for $N = 2$, which is consistent with the fact that the QFI is zero. This implies, of course, that $\omega$ cannot be measured at all, but not that the variance of any unbiased estimator diverges. Rather, the conditions for the QCRB break down: one cannot have an unbiased estimator in an $\epsilon$-interval about the true value $\omega$ if the state is independent of $\omega$: $\langle \hat{\omega} \rangle = \omega$ can only be true at a single point if the lhs is independent of $\omega$, not in a whole finite interval, even if it is arbitrarily small.

## 4.2 QFI for a Bogoliubov transformation followed by a Hamiltonian evolution

In Section 3.3.3 we obtained the QFI associated with a state obtained by applying an operator $\hat{T}_\omega$ followed by an operator $\hat{U}_\omega$. It is expressed via the chain rule (52). This expression takes a much simpler form in the case where $\hat{U}_\omega$ is the evolution operator associated with the $\omega$-dependent Hamiltonian

$$\hat{H}_\omega = \sum_k \epsilon_k(\omega)\hat{n}_k(\omega), \tag{73}$$

describing a system of non-interacting fermions, and $\hat{T}_\omega$ is the Bogoliubov transformation that changes the basis by mapping particles of type $a$ (corresponding to parameter value $\omega_0$) to particles of type $c$ (corresponding to parameter value $\omega$). In (73), $\epsilon_k(\omega)$ are the parameter-dependent single-particle energy eigenvalues, and $\hat{n}_k$ are the occupation number operators. We have, from Eq. (13), the identity

$$|\boldsymbol{n}\rangle_\omega = \hat{T}_\omega |\boldsymbol{n}\rangle_{\omega_0}. \tag{74}$$

Let the initial state be parameter-independent (or, equivalently, taken at a fixed value $\omega_0$ of the parameter). The evolution operator $\hat{U}_\omega = \exp(-i\hat{H}_\omega t)$ is diagonal in the basis $|\boldsymbol{n}\rangle_\omega$, so that

$$
\begin{aligned}
|\psi_\omega(t)\rangle &= \hat{U}_\omega \hat{T}_\omega \left|\psi_{\omega_0}(0)\right\rangle \\
&= e^{-i\hat{H}_\omega t} \sum_{\boldsymbol{n}} \psi_{\boldsymbol{n}} |\boldsymbol{n}\rangle_\omega \\
&= \sum_{\boldsymbol{n}} \psi_{\boldsymbol{n}} e^{-iE_{\boldsymbol{n}}(\omega)t} |\boldsymbol{n}\rangle_\omega \\
&= \sum_{\boldsymbol{n}} \psi_{\boldsymbol{n}}(\omega, t) |\boldsymbol{n}\rangle_\omega,
\end{aligned}
\tag{75}
$$

with $\psi_{\boldsymbol{n}}(\omega, t) = \psi_{\boldsymbol{n}} e^{-iE_{\boldsymbol{n}}(\omega)t}$ and $E_{\boldsymbol{n}}(\omega) = \sum_k \epsilon_k(\omega)n_k$ the total energy of many-body basis state $|\boldsymbol{n}\rangle_\omega$. Thus one can go from $\left|\psi_{\omega_0}(0)\right\rangle$ to a state of the form (40) with two unitaries, one for the change of basis and the other for time evolution.

Our aim is to calculate the QFI of

$$|\psi_\omega\rangle = \sum_{\boldsymbol{n}} e^{-iE_{\boldsymbol{n}}(\omega)t} \psi_{\boldsymbol{n}} |\boldsymbol{n}\rangle_\omega, \tag{76}$$

where $\psi_{\boldsymbol{n}}$ are the coordinates of the initial state $\left|\psi_{\omega_0}(0)\right\rangle$ in the basis $|\boldsymbol{n}\rangle_{\omega_0}$ and thus are independent of $\omega$. We introduce

$$|\gamma_\omega\rangle = \sum_{\boldsymbol{n}} \left(-i\dot{E}_{\boldsymbol{n}}(\omega)t\right) e^{-iE_{\boldsymbol{n}}(\omega)t} \psi_{\boldsymbol{n}} |\boldsymbol{n}\rangle_\omega, \tag{77}$$

$$|\chi_\omega\rangle = \sum_{\boldsymbol{n}} e^{-iE_{\boldsymbol{n}}(\omega)t} \psi_{\boldsymbol{n}} |\dot{\boldsymbol{n}}\rangle_\omega, \tag{78}$$

$$|\varphi_\omega\rangle = \sum_{\boldsymbol{n}} e^{-iE_{\boldsymbol{n}}(\omega)t} \psi_{\boldsymbol{n}} |\boldsymbol{n}\rangle_{\omega_0}, \tag{79}$$

so that $|\dot{\psi}_\omega\rangle = |\gamma_\omega\rangle + |\chi_\omega\rangle$. In terms of $|\gamma_\omega\rangle$ and $|\chi_\omega\rangle$, the QFI Eq. (35) reads

$$
\begin{aligned}
\frac{1}{4}I(|\psi_\omega\rangle, \omega) = {} &\langle\gamma_\omega|\gamma_\omega\rangle + \langle\gamma_\omega|\psi_\omega\rangle^2 + \langle\chi_\omega|\chi_\omega\rangle + \langle\chi_\omega|\psi_\omega\rangle^2 \\
&+ \langle\gamma_\omega|\chi_\omega\rangle + \langle\chi_\omega|\gamma_\omega\rangle + 2\langle\gamma_\omega|\psi_\omega\rangle\langle\chi_\omega|\psi_\omega\rangle.
\end{aligned}
\tag{80}
$$

One then readily gets from (77)

$$\langle \gamma_\omega | \gamma_\omega \rangle + \langle \gamma_\omega | \psi_\omega \rangle^2 = \sum_n |\psi_n|^2 \dot{E}_n^2 t^2 - \left( \sum_n |\psi_n|^2 \dot{E}_n t \right)^2. \tag{81}$$

If we define the diagonal operator $\dot{E} = \sum_n \dot{E}_n |n\rangle\langle n|_{\omega_0}$, this gives

$$\langle \gamma_\omega | \gamma_\omega \rangle + \langle \gamma_\omega | \psi_\omega \rangle^2 = \mathrm{var}(\dot{E}t, \varphi_\omega). \tag{82}$$

Performing the derivative of Eq. (74) with respect to $\omega$, we get

$$|\dot{n}\rangle_\omega = \dot{\hat{T}}_\omega \hat{T}_\omega^\dagger |n\rangle_\omega, \tag{83}$$

and thus

$$|\chi_\omega\rangle = \dot{\hat{T}}_\omega \hat{T}_\omega^\dagger |\psi_\omega\rangle = \dot{\hat{T}}_\omega |\varphi_\omega\rangle. \tag{84}$$

This yields $\langle \chi_\omega | \psi_\omega \rangle = -i \langle \varphi_\omega | \mathcal{H} | \varphi_\omega \rangle$ and thus

$$\langle \chi_\omega | \chi_\omega \rangle + \langle \chi_\omega | \psi_\omega \rangle^2 = \mathrm{var}(\mathcal{H}, \varphi_\omega). \tag{85}$$

Noting that

$$|\dot{\varphi}_\omega\rangle = \sum_n \left( -i\dot{E}_n(\omega)t \right) e^{-iE_n(\omega)t} \psi_n |n\rangle_{\omega_0} = \hat{T}_\omega^\dagger |\gamma_\omega\rangle, \tag{86}$$

the last contribution in Eq. (80) involves the terms

$$\langle \gamma_\omega | \psi_\omega \rangle = i \sum_n |\psi_n|^2 \dot{E}_n t \equiv i \langle \dot{E}t \rangle_{\psi_{\omega_0}} = i \langle \dot{E}t \rangle_{\varphi_\omega}, \tag{87}$$

and $\langle \gamma_\omega | \chi_\omega \rangle$, which from Eqs. (84) and (86) gives

$$\langle \gamma_\omega | \chi_\omega \rangle = i \langle \dot{\varphi}_\omega | \mathcal{H} | \varphi_\omega \rangle = - \langle \dot{E}t\, \mathcal{H} \rangle_{\varphi_\omega}. \tag{88}$$

We obtain

$$\langle \gamma_\omega | \chi_\omega \rangle + \langle \chi_\omega | \gamma_\omega \rangle = - \langle \dot{E}t\, \mathcal{H} \rangle_{\varphi_\omega} - \langle \mathcal{H}\, \dot{E}t \rangle_{\varphi_\omega}, \tag{89}$$

$$\langle \gamma_\omega | \psi_\omega \rangle \langle \chi_\omega | \psi_\omega \rangle = 2 \langle \dot{E}t \rangle_{\varphi_\omega} \langle \mathcal{H} \rangle_{\varphi_\omega}. \tag{90}$$

Summing together all contribution in (80) we get

$$\frac{1}{4} I(|\psi_\omega\rangle, \omega) = \mathrm{var}(\dot{E}t, \varphi_\omega) + \mathrm{var}(\mathcal{H}, \varphi_\omega) - \langle \dot{E}t\, \mathcal{H} \rangle_{\varphi_\omega} - \langle \mathcal{H}\, \dot{E}t \rangle_{\varphi_\omega} + 2 \langle \dot{E}t \rangle_{\varphi_\omega} \langle \mathcal{H} \rangle_{\varphi_\omega}$$
$$= \langle (\dot{E}t - \mathcal{H})^2 \rangle_{\varphi_\omega} - \langle \dot{E}t - \mathcal{H} \rangle_{\varphi_\omega}^2. \tag{91}$$

We thus obtain the very compact expression

$$I(|\psi_\omega\rangle, \omega) = 4\, \mathrm{var}(\dot{E}t - \mathcal{H}, \varphi_\omega), \tag{92}$$

with $\dot{E} = \sum_n \dot{E}_n |n\rangle\langle n|_{\omega_0}$ and

$$_{\omega_0}\langle m| \mathcal{H} |n\rangle_{\omega_0} = i\, _{\omega_0}\langle m| \hat{T}_\omega^\dagger \dot{\hat{T}}_\omega |n\rangle_{\omega_0} = i\, _\omega\langle \dot{m}|n\rangle_\omega. \tag{93}$$

Expression (92) generalizes well-known variance-based formulas [24]. At $t = 0$, state $|\varphi_\omega\rangle$ coincides with $|\psi_{\omega_0}\rangle$ and thus we recover the QFI for a single unitary, Eq. (41).

### 4.3 QFI for a general state

We now put together the results from the previous two subsections and consider the case of a superposition $|\psi_\omega\rangle = \sum_n \psi_n(\omega)|n\rangle_\omega$ of basis states. The QFI is given by Eq. (92), that is, by the variance of $\dot{E}t - \mathcal{H}$ in state $|\varphi_\omega\rangle$. That state is defined by (79), namely,

$$|\varphi_\omega\rangle = \sum_n e^{-iE_n(\omega)t} \psi_n |n\rangle_{\omega_0}, \tag{94}$$

in the basis of kets $|n\rangle_{\omega_0}$. Operator $\mathcal{H}$ corresponds to a Bogoliubov transformation and its action on basis states $|n\rangle_a$ is given by Eq. (70). Since $\omega_0$ is the frequency for type-$a$ particles, we have

$$\mathcal{H}|n\rangle_{\omega_0} = -i \sum_{k<l} (-1)^{\sum_{j=k}^{l-1} n_j} D_{kl}^{(|n_k-n_l|)} |\overline{n}^{k,l}\rangle_{\omega_0}. \tag{95}$$

By linearity, Eqs. (94)–(95) directly give

$$\mathcal{H}|\varphi_\omega\rangle = -i \sum_n \psi_n(\omega,t) \sum_{k<l} (-1)^{\sum_{j=k}^{l-1} n_j} D_{kl}^{(|n_k-n_l|)} |\overline{n}^{k,l}\rangle_{\omega_0}, \tag{96}$$

with $\psi_n(\omega,t) = \psi_n e^{-iE_n(\omega)t}$ Permuting the two sums, we make the change from $\overline{n}^{k,l}$ to $n$ in the sum over $n$. This does not change the term $D_{kl}^{(|n_k-n_l|)}$, while flipping $n_k$ changes the overall sign. This leads to

$$\mathcal{H}|\varphi_\omega\rangle = \sum_n h_n(\omega)|n\rangle_{\omega_0},$$
$$h_n(\omega) = i \sum_{k<l} (-1)^{\sum_{j=k}^{l-1} n_j} D_{kl}^{(|n_k-n_l|)} \psi_{\overline{n}^{k,l}}(\omega,t). \tag{97}$$

The vectors $\mathcal{H}|\varphi_\omega\rangle$ are now both expressed in the same basis $|n\rangle_{\omega_0}$, so that

$$\left(\dot{E}t - \mathcal{H}\right)|\varphi_\omega\rangle = \sum_n \left(\dot{E}_n t \psi_n(\omega,t) - h_n(\omega)\right)|n\rangle_{\omega_0}. \tag{98}$$

We therefore get

$$\mathrm{var}(\dot{E}t - \mathcal{H}, \varphi_\omega) = \sum_n \left|\dot{E}_n t \psi_n(\omega,t) - h_n(\omega)\right|^2 - \left(\sum_n (\dot{E}_n t |\psi_n|^2 - \psi_n^*(\omega,t) h_n(\omega))\right)^2. \tag{99}$$

For a single particle the calculation can be done more easily starting directly from (35). One checks that in that case one gets (99) with $n$ replaced by the index of the single particle states.

## 5 Application to quantum Hall effect

### 5.1 Single-particle quantum Hall physics

We now turn to an application of our results to quantum Hall physics. We consider a two-dimensional system of size $L$ along the $x$-axis and $w$ along the $y$-axis, subjected to a perpendicular constant magnetic field $B$ along the $z$-axis. We choose the coordinate system such that $|y| \leq w/2$, $0 \leq x \leq L$. We denote by $\mathcal{A} = Lw$ the area of the sample. The frequency $\omega = eB/m_{\mathrm{eff}}$ is the cyclotron frequency of charge carriers with effective mass $m_{\mathrm{eff}}$, $l_B = \sqrt{\hbar/(eB)} = \sqrt{\hbar/(m_{\mathrm{eff}}\omega)}$ is the magnetic length, and at the same time the oscillator length associated with frequency $\omega$. We denote with $n_B = 1/(2\pi l_B^2) = B/(h/e)$ the magnetic

flux density (number of flux quanta $\Phi_0 = h/e$ per unit area), and $M = n_B \mathcal{A}$ is the total number of flux quanta.

In the Landau gauge $\mathbf{A} = (-By, 0, 0)$, one can make the Ansatz that the wave function factorizes in $x$ and $y$ direction. Choosing periodic boundary conditions in the $x$-direction results in plane waves in $x$ with wave vector of the form $k_m = m(2\pi/L)$. The effective total Hamiltonian is then given by

$$H = \frac{p_y^2}{2m_{\text{eff}}} + \frac{1}{2} m_{\text{eff}} \omega^2 (y - y_m)^2, \tag{100}$$

where $y_m = k_m l_B^2$ is a shift of the oscillator in the $y$ direction that depends on the quantum number $m$ of the quantization in $x$-direction. The kinetic energy of the plane wave is contained in the $y_m^2$ term. As a consequence, $m$ enters only through the shift $y_m$ in (100) of the origin of the oscillator, and thus energy eigenvalues do not depend on $m$: Landau levels are degenerate. The condition $|y_m| \le w/2$ is equivalent to $|m| \le \mathcal{A}/(4\pi l_B^2) = \mathcal{A} m_{\text{eff}} \omega/(4\pi\hbar) = \Phi/(2\Phi_0)$, which for $\omega$ the cyclotron frequency amounts to $|m| \le M/2$. This is the well-known result that the number $M$ of states per Landau level (LL) $n$, and hence degeneracy of each energy eigenvalue $\hbar\omega_{\text{eff}} n$, is given by the number of flux quanta through the surface. For simplicity we will assume $M$ to be odd, so that $m$ takes the values $m = -(M-1)/2, \ldots, 0, \ldots (M-1)/2$.

The energy eigenstates $|n, m\rangle_\omega$ can be labeled with the two quantum numbers $n, m$. They are conveniently described in the chosen Landau gauge by the wave functions

$$\langle x, y | n, m \rangle_\omega = \frac{e^{ik_m x}}{\sqrt{L}} \chi_{n,m}\left(\frac{y - k_m l_B^2}{l_B}\right), \tag{101}$$

where

$$\chi_{n,m}(\eta) = \frac{\mathcal{N}_n}{\sqrt{l_B}} H_n(\eta) e^{-\eta^2/2}, \tag{102}$$

is the usual harmonic-oscillator wave function in terms of the Hermite polynomial $H_n(\eta)$, while

$$\mathcal{N}_n = \frac{1}{\sqrt{2^n \sqrt{\pi} n!}}, \tag{103}$$

is a normalization factor.

## 5.2 Wave function overlaps

In order to calculate the QFI using Eq. (72), we first need to obtain the derivative of the matrix $R$ of overlaps. We calculate the overlap between states $|n, m\rangle_{\omega_0 + \delta\omega}$, where the frequency differs from $\omega_0$ by an infinitesimal amount $\delta\omega$, and states $|n', m'\rangle_{\omega_0}$ at some fixed frequency $\omega_0$. At first order,

$$_{\omega_0}\langle n', m' | n, m \rangle_{\omega_0 + \delta\omega} \simeq \delta_{n,n'} \delta_{m,m'} + {}_{\omega_0}\langle n', m' | \partial_{\omega_0} | n, m \rangle_{\omega_0} \delta\omega. \tag{104}$$

The calculation of the first-order term is detailed in Appendix B. We get the final expression

$$_{\omega_0}\langle n', m' | n, m \rangle_{\omega_0 + \delta\omega} \simeq \delta_{n,n'} \delta_{m,m'} + \left[ \frac{k_m l_B}{\sqrt{2}\omega_0} \left( \sqrt{n} \delta_{n',n-1} - \sqrt{n+1} \delta_{n',n+1} \right) \right.$$
$$\left. + \frac{1}{4\omega_0} \left( \sqrt{n(n-1)} \delta_{n',n-2} - \sqrt{(n+2)(n+1)} \delta_{n',n+2} \right) \right] \delta_{m',m} \delta\omega. \tag{105}$$

An alternative way of deriving this quantity is to start from the Hutchisson formula [27] for $R_{n'n} = {}_{\omega_0}\langle n'|n\rangle_\omega$, given by

$$
R_{n'n} = \sqrt{2^{-(n+n')}qn!n'!}(-1)^n e^{-\frac{1}{4}\gamma^2 p} \sum_{r=0}^{\min(n,n')} \frac{(-2q)^r}{r!} \sum_{s=0}^{\lfloor(n-r)/2\rfloor} \frac{(\gamma p)^{n-r-2s}}{(n-r-2s)!} \frac{x^s}{s!}
$$
$$
\times \sum_{t=0}^{\lfloor(n'-r)/2\rfloor} \frac{(\gamma q)^{n'-r-2t}}{(n'-r-2t)!} \frac{(-x)^t}{t!} , \tag{106}
$$

with $\gamma = k_m(l_B^2(\omega) - l_B^2(\omega_0))/l_B(\omega)$, $x = (\omega - \omega_0)/(\omega + \omega_0)$, $q = 2(\omega\omega_0)^{1/2}/(\omega + \omega_0)$, and $p = 2\omega_0/(\omega + \omega_0)$ (we took the formula of [27] with $\nu' = \omega_0$ and $\nu'' = \omega$). This approach is more cumbersome. As a check, we show in Appendix C that a first-order expansion of (106) around $\omega = \omega_0$ allows us to recover the result (105).

## 5.3 Gauge choice and occupation numbers

We now consider the basis state $|\mathbf{n}\rangle_\omega$ where $N$ particles fill the lowest available energy levels.

The filling factor $\nu = N/M$, with $M$ the number of states per LL, determines how many LLs are occupied. The largest integer smaller than $\nu$ is denoted by $f$. In an infinitely extended sample without additional potentials ("ideal sample"), it determines the last fully occupied LL. The last LL is occupied by only $\bar{m}$ particles, with $N = Mf + \bar{m}$.

In an ideal sample all single-electron states with the same $n$ are degenerate in energy, and the larger the value of $k_m$ the larger the sensitivity of these states to a change of magnetic field. Indeed $k_m$ determines how quickly the wavefunctions oscillate, and hence how sensitive they are to a change of $l_B$ with $B$. Importantly, while the different values that $k_m$ can take are to a certain degree arbitrary in the ideal system of flat Landau levels in the absence of an electrostatic potential, we consider in the following a smoothly varying confinement potential that lifts the degeneracy at the different values of $k_m$. The absolute values of $k_m$ therefore matter for the QFI, as the QFI depends on the energy eigenstate considered.

As we have just mentioned, in a real sample, the degeneracy in energy is broken by the electrostatic potential, which takes into account both smooth disorder and a possible confinement potential. The order in which the LL states are occupied is nontrivial, and this can influence the QFI. Here, we omit disorder and consider a smooth confinement potential. In a sensor based on a two-dimensional electron gas (2DEG), confined electrostatically by metallic electrodes at a substantial distance from the 2DEG, on the order of $100\,\mathrm{nm}$, the confining potential varies on a length scale typically much larger than $l_B$ for a magnetic field on the order of $1T$. The additional potential hardly modifies the electron wave functions in this case and hence just leads to a shift of the energy eigenvalues corresponding to the value of the potential where the energy eigenstate is localized. By symmetry, one can expect the minimum of the confining potential to lie at the center of the sample, where it can be approximated by a slow-varying potential. The shorter the sample in a given direction, the stronger a variation of the confining potential in that direction, and hence, the larger the additional potential energy. This implies that the lowest-energy single electron states to be populated are oscillator states extended in the *largest* direction of the sample, where the potential energy due to the confining potential grows more slowly. This situation is naturally taken care of in the Landau gauge: $A_x = -By$ for $w \gg L$, but $A_y = Bx$ for $L \gg w$ (with the other components of $\mathbf{A}$ equal zero). For $w \simeq L$, a symmetric Landau gauge $\mathbf{A} = B(-y, x, 0)$ is most appropriate and leads to axial-symmetric wave functions and conservation of angular momentum instead of linear momentum. But since the additional potential is, for $w = L$, also symmetric under $w \leftrightarrow L$, both $A_x = -By$ and $A_y = Bx$ should lead to the same result as the symmetric gauge in this case.

In addition to this confining potential, there is typically also a disorder potential in a real sample. Disorder arises from impurities or dopants that are in general relatively far from the 2DEG as well, and hence lead to a random potential that varies slowly over the sample. Energy eigenstates are then localized at the minimum of this potential and filled in order of increasing energy, like puddles. The quantum number $m$ ceases to be a good quantum number and is replaced by a quantum number that labels the position where the oscillator state corresponding to the Landau levels are localized. This implies a QFI that varies randomly from sample to sample, with a statistics that is, however, beyond the scope of the paper. In the following we restrict ourselves to a clean sample with only a confining potential that breaks the degeneracy in energy of the LLs.

### 5.4 Quantum Fisher information for the $N$-particle quantum Hall effect

We shall now focus on the case $w \gg L$ with $A_x = -By$. The QFI is given by (72), which involves a sum over all pairs of labels $(k, l)$ with $k < l$, so that only labels such that the occupation number differs by 1 contribute. If only the lowest levels are filled, the sum runs over all pairs $k < l$ such that level $k$ is occupied and level $l$ is empty. In the present context of the quantum Hall effect, each label $k$ has to be replaced by two quantum numbers $(n, m)$; the QFI is thus a sum over all pairs of contributions such that $(n, m)$ is occupied and $(n', m')$ is empty. The summand is the derivative of the overlap between a level $(n, m)$ and a level $(n', m')$, obtained from Eq. (105); it reads

$$\dot{R}_{n',m';n,m} = \left[ \frac{k_m l_B \left( \sqrt{n}\,\delta_{n',n-1} - \sqrt{n+1}\,\delta_{n',n+1} \right)}{\sqrt{2}\,\omega_0} \right.$$
$$\left. + \frac{\sqrt{n(n-1)}\,\delta_{n',n-2} - \sqrt{(n+2)(n+1)}\,\delta_{n',n+2}}{4\omega_0} \right] \delta_{m'm} \,. \tag{107}$$

Recall from Sect. 5.1 that $m = -\frac{M-1}{2}, \ldots, 0, \ldots \frac{M-1}{2}$ and $N = Mf + \bar{m}$. The fully filled LLs are labelled $n = 0, .., f-1$, while the last LL $n = f$ contains $\bar{m}$ particles filling states $\left| n, -\frac{M-1}{2} + i \right\rangle$ with $0 \le i \le \bar{m} - 1$. From the $\delta_{m'm}$ in (107) only pairs with $m = m'$ contribute, therefore we sum over pairs $(n, m)$ and $(n', m)$ with $n < n'$ and with $(n, m)$ occupied and $(n', m)$ empty. If $m = -\frac{M-1}{2} + i$ with $0 \le i \le \bar{m} - 1$, the occupied states are $n = 0, \ldots, f$ and the empty ones are $n' \ge f + 1$. If $m \ge -\frac{M-1}{2} + \bar{m}$, the occupied states are $n = 0, \ldots, f-1$ and the empty ones are $n' \ge f$. Thus, the QFI (72) can be expressed as

$$I(|\boldsymbol{n}\rangle_\omega, \omega) = 4 \left( \sum_{m=-\frac{M-1}{2}}^{\bar{m}-\frac{M+1}{2}} \sum_{n=0}^{f} \sum_{n'=f+1}^{\infty} + \sum_{m=\bar{m}-\frac{M-1}{2}}^{\frac{M-1}{2}} \sum_{n=0}^{f-1} \sum_{n'=f}^{\infty} \right) |\dot{R}_{n',m;n,m}|^2 \,. \tag{108}$$

Note that from the delta function in (107) only pairs with $m = m'$ contribute. Only terms with $n < n'$ contribute to the sum, so that only terms $\delta_{n',n+1}$ and $\delta_{n',n+2}$ survive. Equation (108) reduces to

$$I(|\boldsymbol{n}\rangle_\omega, \omega) = 4 \left( \sum_{m=-\frac{M-1}{2}}^{\bar{m}-\frac{M+1}{2}} \sum_{n=0}^{f} \sum_{n'=f+1}^{\infty} + \sum_{m=\bar{m}-\frac{M-1}{2}}^{\frac{M-1}{2}} \sum_{n=0}^{f-1} \sum_{n'=f}^{\infty} \right)$$
$$\times \left( \frac{(k_m l_B)^2 (n+1)}{2\omega^2} \delta_{n',n+1} + \frac{(n+2)(n+1)}{16\omega^2} \delta_{n',n+2} \right), \tag{109}$$

where we have set $\omega_0 = \omega$. Only pairs $n, n'$ differing by 1 or 2 units contribute, thus (109) becomes

$$
I(|\boldsymbol{n}\rangle_\omega, \omega) = \frac{1}{\omega^2} \left[ \sum_{m=-(M-1)/2}^{\bar{m}-(M+1)/2} \left( 2(k_m l_B)^2 (f+1) + \frac{1}{2}(f+1)^2 \right) \right.
$$
$$
\left. + \sum_{m=\bar{m}-(M-1)/2}^{(M-1)/2} \left( 2(k_m l_B)^2 f + \frac{1}{2} f^2 \right) \right]. \tag{110}
$$

Replacing $k_m$ by its value $2\pi m/L$, one can perform the sum over $m$. This leads to the final expression

$$
I(|\boldsymbol{n}\rangle_\omega, \omega) = \frac{1}{6L^2 \omega^2} \Big( 3L^2(-f(f+1)M + 2fN + N)
$$
$$
- 4\pi^2 l_B^2 \big( 2f(f+1)(2f+1)M^3 + 6(2f+1)MN^2
$$
$$
- 3N(2fM+M)^2 - 4N^3 + N \big) \Big). \tag{111}
$$

The first line is independent of $L$ and the geometry of the sample. The remaining terms both depend on $B$ and $L$. Since $\omega$ and $B$ are linearly related, error propagation leads to the same relative minimal standard deviation of an unbiased estimator $\sigma(\hat{B})$ of $B$ as for $\omega$. Together with Eq. (32) we obtain

$$
\sigma(\hat{B}) \geq \frac{m_{\text{eff}}/e}{\sqrt{M_e I(|\boldsymbol{n}\rangle_\omega, \omega)}}, \tag{112}
$$

where $M_e$ is the number of independent measurements. A necessary condition for the application of the formula is that the very description of the system in terms of harmonic oscillators is adequate. This implies that the magnetic field must not be too weak, i.e. $l_B \ll w$, which sets a lower bound on $B$ for given $w$, $B \gg \hbar/(ew^2)$ with numerical values $B[T] \gg 6.58 \times 10^{-16}/(w[\text{m}])^2$. Conversely, for given $B$ the formula implies a minimal sensor size of $w > 25.7\,\text{nm}/(\sqrt{B[\text{T}]})$. Secondly, we recall that we assumed $w \gg L$. In the opposite case, $w$ and $L$ should be exchanged. As explained above, symmetry under exchange of $w$ and $L$ is not to be expected in a sensor where the confining potential breaks that symmetry.

The most interesting regime corresponds to $N \gg 1$. In a realistic sample, the areal density $n_{2D} = N/(Lw)$ of the electrons is fixed. In this case, for increasing $N$, $w$ or $L$ must increase as well, and with them $M \propto Lw \propto N$. Hence, in the limit of large $N$ one should replace $M$ by its value $(N - \bar{m})/f$. Suppose $w = \mu N^\lambda l_B$ and $L = \nu N^{1-\lambda} l_B$ with $1/2 \leq \lambda \leq 1$ to ensure that $w \gg L$ for large $N$. For $\lambda > 1/2$, one always has $w \gg L$ for $N \to \infty$, whereas for $\lambda = 1/2$, $w/L = \mu/\nu$ is fixed but can be large. Then the leading term in $N$ of the QFI becomes

$$
I(|\boldsymbol{n}\rangle_\omega, \omega) \simeq \frac{2\pi^2}{3f^2 \omega^2 \nu^2} N^{1+2\lambda}, \tag{113}
$$

and signals faster than "Heisenberg scaling" of the QFI (meaning $I(|\boldsymbol{n}\rangle_\omega, \omega) \propto N^2$ [28]) for $\lambda > 1/2$. The fastest possible scaling, $I(|\boldsymbol{n}\rangle_\omega, \omega) \propto N^3$ can be achieved in the limit of fixed $L$ and correspondingly $w \propto N$, i.e. in the limit of a strip-like sensor.

The origin of this "super-Heisenberg scaling" can be traced back to the $\sum_m m^2$ in (110) with bounds $\sim M \sim N$ which gives a scaling $\propto N^3$. It arises from occupying high-momentum states in $x$-direction, as is required for Fermions by the Pauli principle. Since $k_m$ determines also the shift $y_m = k_m l_B^2$ of the harmonic oscillators in $y$ direction, large values of $k_m$ lead to correspondingly large sensitivity to a change of $l_B$ and hence of $B$. Interestingly, if the kinetic energy in $x$-direction had a power-law scaling with $k_x$ with a different power, also the scaling

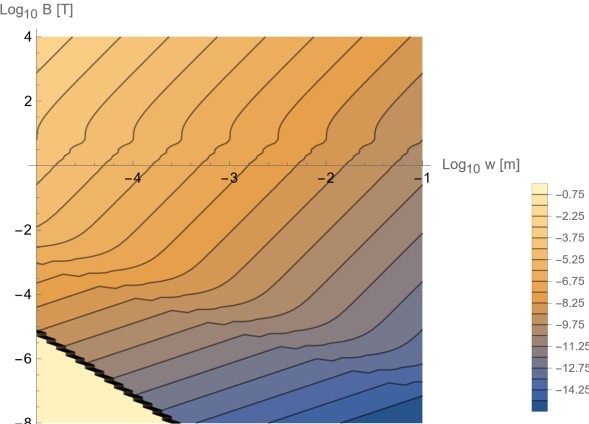

Figure 1: Logarithmic contour plot of the minimal standard deviation, i.e. $\log_{10} \sigma(\hat{B})$, of an unbiased estimate $\hat{B}$ of the magnetic field $B$ based on the quantum Hall effect, Eqs. (112) and (111) for a single readout, $M_e = 1$, as function of the width $w$ of the sensor and magnetic field $B$ for $10^{-8} \leq B[T] \leq 10^2$. The minimal error is plotted only where the theory is applicable, $w > L, l_B(B)$. The length is set to $L = 1\mu$ m, and $w \geq 10^{-5}$ m.

of the sensitivity with $N$ would change, with higher powers being favorable. Numerically, using typical parameter values for Gallium arsenide ($m_{\text{eff}} \simeq 0.068 m_e$, $n_{2D} \simeq 1.0 \times 10^{15}/\text{m}^2$) and a magnetic field of 1 T, the minimal predicted error is on the order of $6.2 \times 10^{-11}$ T for a single measurement with a sensor of size $w = 1$ cm, $L = 1$ mm. The number of information-carrying measurements per second is determined by the bandwidth of the interrogation scheme. That bandwidth is ultimately limited by the cyclotron frequency, and hence the number of measurements in 1 s cannot be greater than $M_e \sim 1/(\omega \cdot 1s) \simeq 10^{12}$ at 1 T. A more conservative bandwidth of 10 G Hz yields a bound on the achievable sensitivity on the order of $10^{-16}$ T/Hz$^{1/2}$, to be compared with $100$ nT/Hz$^{1/2}$ sensitivity of a silicon-based commercially available sensor [1], or another one with $0.4 \mu$T sensitivity at fields up to about mT with a chip of linear size $\sim$mm [29].

In Fig. 1 we plot the minimal estimation error $\sigma(B)$ as function of $B$ and $w$ in the parameter ranges where $l_B(B) < w$ is satisfied, which is everywhere outside the yellow left lower corner, whose boundary indeed scales as $w \propto 1/\sqrt{B}$. The length is fixed in this plot, i.e. $\lambda = 1$ in Eq. (113). The expected scaling $\sigma(B) \propto w^{-3/2}$ is reached for sufficiently large $w$, as can be seen e.g. for $B = 1$ T. It should be kept in mind that *i.)* we considered the case of zero temperature and neglected decoherence, and *ii.)* the QCRB provides an idealized lower bound on the error that can rarely be achieved in practice due to additional technical noise and other imperfections. Nevertheless, the QCRB (112) constitutes an important benchmark that allows one to understand what sensitivity is possible in principle as function of $B$ and the size of the sensor.

## 6  Conclusion

In summary, we have derived analytical expressions for the quantum Fisher information (QFI) that determines the smallest possible fluctuations of an unbiased estimator of a parameter encoded in an arbitrary pure quantum state of a fermionic many-body system via three different types of unitary transformations. In the case of two concatenated unitaries we obtained a simple chain rule for the QFI, Eq. (52), that simplifies further for parameters coded through a

Bogoliubov transformation (61), for a many-body basis state (71), or a Hamiltonian time evolution paired with a modification of the single-particle energy eigenstates (92). In the latter case, a variance of a Hermitian generator naturally arises just as for a single unitary evolution, albeit taken in an intermediate state. We applied the general results to the quantum Hall effect in the ground state of non-interacting electrons and calculated the smallest possible standard deviation of an unbiased estimator of the magnetic field. We found a scaling of the sensitivity (standard deviation) with which the magnetic field can be measured as $1/\sqrt{N^{1+2\lambda}}$, where $\lambda \in [1/2, 1]$ controls the scaling of the width and length of the sensor with the number of electrons.

For any $\lambda > 1/2$ this corresponds to a "super-Heisenberg" scaling of the sensitivity, a term that is somewhat controversial. In [30] it was shown in the context of quantum metrology of a parameter that drives a many-body system through a continuous quantum critical point, with an energy gap which vanishes polynomially with $N$ at the critical point, that no faster than Heisenberg scaling can be achieved without violating the assumption of adiabaticity if the time needed for preparing the initial state through adiabatic driving is taken into account. Our scenario is not of that type, however. We do not consider a phase transition with a closing gap, nor a time-dependent parameter in the present case of the quantum Hall effect. The QFI that we calculate, Eq. (113), does hence not contain time. Its value merely reflects the dependence of the single-particle energy eigenstates on the cyclotron frequency. The initial state of the sensor can be reached for any value of the magnetic field by cooling the sample close to the ground state. One can nevertheless prepare the initial state starting from a different value of the magnetic field by modifying the magnetic field as a function of time; however, we still make the assumption that degeneracy of the energy-eigenstates is lifted at least by the confinement potential. In real samples, a disorder potential lifts the degeneracy additionally, and for generic values of the magnetic field one expects no closed gaps, regardless the number of electrons. Therefore, even in such a scenario, where the magnetic field is driven as function of time we do not expect that a slowing down of the rate of change of the parameter is required with increasing $N$ in order to remain adiabatic over an infinitesimal change of the parameter.

The "super-Heisenberg scaling" has its physical origin in the occupation of high-energy momentum states, as required by the Pauli principle, which lead to large spatial displacements of the energy eigenstates corresponding to the Landau-levels, proportional to the momentum and the magnetic length squared. The large momenta hence translate to high sensitivity to changes of the magnetic length and as a consequence of the magnetic field. It should be kept in mind, however, that the analysis is highly idealized: zero temperature was assumed, and all decoherence effects as well as technical noise are neglected. Future work will have to show how robust the large sensitivities are, and how they change when using different materials.

## Acknowledgments

DB thanks OG and the University Paris-Saclay for hospitality for a stay during which part of this work was done.

## A  Alternative proof of (57)

Here we sketch an alternative proof of (57). For brevity we define $\hat{H} = i\hat{S}$, and a $2M \times 2M$ matrix $H = iS/2$, i.e. $\hat{H} = \alpha H \alpha^t$. One then shows in the fermionic case by direct calculation that

$$[\hat{H}, \dot{\hat{H}}] = 2\alpha(H, \dot{H})\alpha^t, \tag{A.1}$$

where the commutator-like bilinear form $(A, B)$ of two operators is defined as

$$(A, B) \equiv A \Xi B - B \Xi A, \tag{A.2}$$

with $\Xi$ defined in Eq. (6). Eq. (A.1) generalizes to higher order commutators $[A, B]_n$ defined recursively through $[A, B]_{n+1} = [A, [A, B]_n]$ and $[A, B]_0 = B$, and correspondingly $(A, B)_{n+1} = (A, (A, B)_n)$ and $(A, B)_0 = B$:

$$[\hat{H}, \dot{\hat{H}}]_n = 2^n \alpha (H, \dot{H})_n \alpha^t . \tag{A.3}$$

Next, one can write the derivative of an exponential of a parameter dependent operator alternatively as

$$\frac{\partial}{\partial \omega} e^{\beta H(\omega)} = \int_0^\beta ds \, e^{(\beta-s)H(\omega)} \frac{\partial H(\omega)}{\partial \omega} e^{sH(\omega)} , \tag{A.4}$$

where $\beta$ is an arbitrary real number. The simplest proof of (A.4) follows the lines of [31] by showing that both sides of the equation satisfy the first-order differential equation

$$\frac{\partial F}{\partial \beta} - H(\omega)F(\beta) = \frac{\partial H(\omega)}{\partial \omega} e^{\beta H(\omega)} , \tag{A.5}$$

together with $F(0) = 0$, which fixes the solution uniquely. Next one checks the identities $e^{A\Xi}Be^{-\Xi A} = \sum_{n=0}^\infty \frac{1}{n!}(A, B)_n$ and $e^A B e^{-A} = \sum_{n=0}^\infty \frac{1}{n!}[A, B]_n$. With this we have

$$\mathcal{H} = -i\hat{T}^\dagger \dot{\hat{T}} = (-i) \int_0^1 du \, e^{-iu\hat{S}} i \frac{\partial \hat{S}}{\partial \omega} e^{-iu\hat{S}} \tag{A.6}$$

$$= \int_0^1 du \sum_{n=0}^\infty \frac{1}{n!} [-iu\hat{S}, \dot{\hat{S}}]_n \tag{A.7}$$

$$= \frac{\alpha}{2} \left[ \int_0^1 du \sum_{n=0}^\infty \frac{(-iu)^n}{n!} (S, \dot{S})_n \right] \alpha^t \tag{A.8}$$

$$= \frac{\alpha}{2} \int_0^1 du \, e^{-iuS\Xi} \dot{S} e^{iu\Xi S} \alpha^t \tag{A.9}$$

$$= \frac{1}{2} \alpha \tilde{\Omega} \alpha^t , \tag{A.10}$$

with $\tilde{\Omega}$ given by (58), which completes the proof.

## B  Derivation of wavefunction overlaps

We start from Eq. (104). Instead of varying the frequency, it is more convenient to make the change of variables to the magnetic length $l \equiv l_B = \sqrt{\hbar/(m_{\text{eff}}\omega)}$ (for ease of notation, in the present Appendix we denote the magnetic length just by $l$). Equation (104) becomes

$$_{\omega_0}\langle n', m' | n, m \rangle_{\omega_0+\delta\omega} \simeq \delta_{n,n'} \delta_{m,m'} + {}_l\langle n', m' | \partial_l | n, m \rangle_l \frac{dl}{d\omega_0} \delta\omega , \tag{B.1}$$

and $dl/d\omega_0 = -(1/2)l/\omega_0$. The matrix element

$$_l\langle n', m' | \partial_l | n, m \rangle_l = \int dx dy \, {}_l\langle n', m' | x, y \rangle \partial_l \langle x, y | n, m \rangle_l , \tag{B.2}$$

can now be calculated from the explicit expression (101) of the wave functions. In particular (101) gives

$$\partial_l \langle x, y | n, m \rangle_l = \frac{e^{ik_m x}}{\sqrt{L}} \partial_l \chi_{n,m}(\eta),$$  (B.3)

with

$$\eta = \frac{y}{l} - k_m l.$$  (B.4)

Integration over $x$ yields a $\delta_{m,m'}$ coefficient. The matrix element (B.2) becomes

$$_l\langle n', m' | \partial_l | n, m \rangle_l = \delta_{m,m'} l \int d\eta \, \chi_{n',m'}(\eta) \partial_l \chi_{n,m}(\eta).$$  (B.5)

Using

$$\frac{d\eta}{dl} = -\frac{y}{l^2} - k_m = -\frac{\eta}{l} - 2k_m,$$  (B.6)

we get

$$\partial_l \chi_{n,m}(\eta) = -\frac{\chi_{n,m}(\eta)}{2l} + \frac{\mathcal{N}_n}{\sqrt{l}} \frac{d\eta}{dl} \partial_\eta \left[ H_n(\eta) e^{-\eta^2/2} \right]$$
$$= -\frac{\chi_{n,m}(\eta)}{2l} - \frac{\mathcal{N}_n}{\sqrt{l}} \left( \frac{\eta}{l} + 2k_m \right) \left[ \partial_\eta H_n(\eta) - \eta H_n(\eta) \right] e^{-\eta^2/2}.$$  (B.7)

One thus obtains for the matrix element (B.2)

$$_l\langle n', m' | \partial_l | n, m \rangle_l$$

$$= -\frac{1}{2l} \delta_{n,n'} \delta_{m,m'} - \delta_{m,m'} \mathcal{N}_n \mathcal{N}_{n'} \int d\eta \, e^{-\eta^2} H_{n'}(\eta) \left( \frac{\eta}{l} + 2k_m \right) \left[ \partial_\eta H_n(\eta) - \eta H_n(\eta) \right]$$

$$= -\frac{1}{2l} \delta_{n,n'} \delta_{m,m'} - \delta_{m,m'} \mathcal{N}_n \mathcal{N}_{n'} \int d\eta \, e^{-\eta^2} H_{n'}(\eta) \left( \frac{\eta}{l} + 2k_m \right) \left[ 2n H_{n-1}(\eta) - \eta H_n(\eta) \right]$$

$$= -\frac{1}{2l} \delta_{n,n'} \delta_{m,m'}$$
$$- \delta_{m,m'} \mathcal{N}_n \mathcal{N}_{n'} \int d\eta \, e^{-\eta^2} H_{n'}(\eta) \left[ 4k_m n H_{n-1}(\eta) + \frac{2n}{l} \eta H_{n-1}(\eta) - 2k_m \eta H_n(\eta) - \frac{\eta^2}{l} H_n(\eta) \right],$$  (B.8)

where we have used $\partial_\eta H_n(\eta) = 2n H_{n-1}(\eta)$ [32]. These integrals can be evaluated if we express $\eta$ and $\eta^2$ in terms of Hermite polynomials,

$$H_1(\eta) = 2\eta, \qquad H_2(\eta) = 2(2\eta^2 - 1) \Leftrightarrow \eta^2 = \frac{1}{2} + \frac{1}{4} H_2(\eta),$$  (B.9)

so that (B.8) becomes

$$_l\langle n', m' | \partial_l | n, m \rangle_l = -\frac{1}{2l} \delta_{n,n'} \delta_{m,m'} - \delta_{m,m'} \mathcal{N}_n \mathcal{N}_{n'} \int d\eta \, e^{-\eta^2} H_{n'}(\eta) \big[ 4k_m n H_{n-1}(\eta)$$  (B.10)

$$+ \frac{n}{l} H_1(\eta) H_{n-1}(\eta) - k_m H_1(\eta) H_n(\eta) - \frac{1}{2l} H_n(\eta) - \frac{1}{4l} H_2(\eta) H_n(\eta) \big].$$

We can now use the identity [32]

$$\int_{-\infty}^{\infty} d\eta \, e^{-\eta^2} H_{n'}(\eta) H_m(\eta) H_n(\eta) = \frac{2^{(m+n+n')/2} \sqrt{\pi} n'! n! m!}{(s-n')!(s-n)!(s-m)!},$$  (B.11)

where $s = \frac{1}{2}(n + n' + m)$ (note that $s$ is an integer due to the parity of the Hermite polynomials – indeed, for an odd integer value of $n + n' + m$ the integrand in (B.11) is an odd function and thus the integral vanishes). We also make use of the orthogonality relation of the Hermite polynomials

$$\int_{-\infty}^{\infty} d\eta\, e^{-\eta^2} H_{n'}(\eta) H_n(\eta) = \sqrt{\pi} 2^n n! \delta_{n',n}, \tag{B.12}$$

so that the matrix element (B.10) becomes

$$\begin{aligned}
{}_l\langle n', m' | \partial_l | n, m \rangle_l = {}&-2\sqrt{2n} k_m \delta_{n',n-1} \delta_{m',m} \\
&- \delta_{m',m} \mathcal{N}_n \mathcal{N}_{n'} \left[ \frac{n}{l} \frac{2^{(n+n')/2} \sqrt{\pi} n'! (n-1)!}{\left(\frac{n'-n}{2}+1\right)! \left(\frac{n-n'}{2}\right)! \left(\frac{n+n'}{2}-1\right)!} \Pi(n+n') \right.\\
&\qquad\qquad - k_m \frac{2^{(n+n'+1)/2} \sqrt{\pi} n'! n!}{\left(\frac{n+1-n'}{2}\right)! \left(\frac{n'+1-n}{2}\right)! \left(\frac{n+n'-1}{2}\right)!} \Pi(n+n'+1) \\
&\qquad\qquad \left. - \frac{1}{4l} \frac{2^{2+(n+n')/2} \sqrt{\pi} n'! n!}{\left(\frac{n-n'}{2}+1\right)! \left(\frac{n'-n}{2}+1\right)! \left(\frac{n+n'}{2}-1\right)!} \right] \Pi(n+n'),
\end{aligned} \tag{B.13}$$

where we have introduced a parity function

$$\Pi(m) = \begin{cases} 1, & \text{for } m \text{ even,} \\ 0, & \text{for } m \text{ odd,} \end{cases} \tag{B.14}$$

to take into account the different cases for which the integral (B.11) vanishes. Further simplifications can be obtained by noticing that the factorials in the denominators of (B.13) involve both $n'-n$ and $n-n'$ in each term. Since the factorial of a negative number is infinite, we get the following simplifications

$$\frac{\Pi(n+n')}{\left(\frac{n'-n}{2}+1\right)! \left(\frac{n-n'}{2}\right)!} = \delta_{n',n-2} + \delta_{n',n}, \tag{B.15}$$

$$\frac{\Pi(n+n'+1)}{\left(\frac{n+1-n'}{2}\right)! \left(\frac{n'+1-n}{2}\right)!} = \delta_{n',n-1} + \delta_{n',n+1}, \tag{B.16}$$

$$\frac{\Pi(n+n')}{\left(\frac{n-n'}{2}+1\right)! \left(\frac{n'-n}{2}+1\right)!} = \frac{1}{2} \delta_{n',n-2} + \delta_{n',n} + \frac{1}{2} \delta_{n',n+2}. \tag{B.17}$$

Replacing the normalization factors $\mathcal{N}_n \mathcal{N}_{n'}$ by their value (103), Eq. (B.13) then reduces to

$$\begin{aligned}
{}_l\langle n', m' | \partial_l | n, m \rangle_l = {}&-2\sqrt{2n} k_m \delta_{n',n-1} \delta_{m',m} - \delta_{m',m} \sqrt{n'! n!} \left[ \frac{1}{2l} \frac{1}{\left(\frac{n+n'}{2}-1\right)!} \left( \delta_{n',n-2} - \delta_{n',n+2} \right) \right. \\
&\left. - \sqrt{2} k_m \frac{1}{\left(\frac{n+n'-1}{2}\right)!} \left( \delta_{n',n-1} + \delta_{n',n+1} \right) \right],
\end{aligned} \tag{B.18}$$

which further simplifies to

$$\begin{aligned}
{}_l\langle n', m' | \partial_l | n, m \rangle_l = {}&-\sqrt{2} k_m \left( \sqrt{n} \delta_{n',n-1} - \sqrt{n+1} \delta_{n',n+1} \right) \delta_{m',m} \\
&- \frac{1}{2l} \left( \sqrt{n(n-1)} \delta_{n',n-2} - \sqrt{(n+2)(n+1)} \delta_{n',n+2} \right) \delta_{m',m}.
\end{aligned} \tag{B.19}$$

Expanding expression (B.1) to linear order in $\delta\omega$ we directly get Eq. (105).

## C Alternative derivation of (105)

We start from the Hutchisson formula for the overlaps $R_{nn'} = {}_{\omega_0}\langle n|n'\rangle_\omega$. It is given by [27] with $\nu' = \omega_0$ and $\nu'' = \omega$, and reads

$$R_{nn'} = \sqrt{2^{-(n+n')}qn!n'!}(-1)^{n'}e^{-\frac{1}{4}\gamma^2 p}\sum_{r=0}^{\min(n,n')}\frac{(-2q)^r}{r!}\sum_{s=0}^{\lfloor(n'-r)/2\rfloor}\frac{(\gamma p)^{n'-r-2s}}{(n'-r-2s)!}\frac{x^s}{s!}$$
$$\times \sum_{t=0}^{\lfloor(n-r)/2\rfloor}\frac{(\gamma q)^{n-r-2t}}{(n-r-2t)!}\frac{(-x)^t}{t!}, \tag{C.1}$$

with $\gamma = \sqrt{\omega m_{\text{eff}}/\hbar}d = d/l_B(\omega)$, $d = k_m(l_B^2(\omega) - l_B^2(\omega_0))$, $x = (\omega - \omega_0)/(\omega + \omega_0)$, $q = 2(\omega\omega_0)^{1/2}/(\omega + \omega_0)$, and $p = 2\omega_0/(\omega + \omega_0)$.

We want to calculate the first derivative of $R_{nn'}$ with respect to $\omega$, taken at $\omega = \omega_0$. In that limit we have $(\gamma^2 p)' = q' = 0$. Contributions to the derivative will therefore come from derivatives of terms of the form $(\gamma p)^a(\gamma q)^b x^c$ with $a, b, c \geq 0$, that is,

$$[(\gamma p)^a(\gamma q)^b x^c]' = a(\gamma p)^{a-1}(\gamma p)'(\gamma q)^b x^c + (\gamma p)^a b(\gamma q)^{b-1}(\gamma q)' x^c + (\gamma p)^a(\gamma q)^b c x^{c-1}x'. \tag{C.2}$$

In the limit $\omega = \omega_0$ we have $\gamma = x = 0$, so that only terms with an exponent 0 in (C.2) can yield a nonzero contribution. Equation (C.2) reduces to $(\gamma p)' = -k_m/\omega_0^{3/2}$ for $a = 1, b = c = 0$, to $(\gamma q)' = -k_m/\omega_0^{3/2}$ for $b = 1, a = c = 0$, and to $x' = 1/(2\omega_0)$ for $c = 1, a = b = 0$. The first case corresponds to $s = t = 0$, $r = n' - 1 = n$. The second case gives $s = t = 0$ and $r = n' = n - 1$. The third case leads to either $s = 0, t = 1, r = n' = n - 1$ or $s = 1, t = 0, r = n' - 1 = n$. Gathering all contributions together we exactly get the first-order term of Eq. (105).

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
