# Peer review of "Quantum parameter estimation with many-body fermionic systems and application to the Hall effect"

_SciPost Physics, doi:SciPost Phys. 16, 085 (2024)_

## Round 1 · Referee Report · Anonymous (Referee 1) · 2023-8-4

Strengths

-detailed derivations
-general formula for quantum Fisher information for fermionic systems with Gaussian parameter encoding
-application to quantum Hall system

Weaknesses

-motivations for some calculations missing
-unclear definition of Heisenberg scaling

Report

The authors Olivier Giraud, Mark-Oliver Goerbig, and Daniel Braun study in their manuscript ''Quantum parameter estimation with many-body fermionic systems and application to the Hall effect'' the quantum Fisher information (QFI) for a generic fermionic quantum many-body state depending on a single parameter. The single parameter is encoded into the wave function by the application of a Gaussian unitary while they distinguish between three different kinds: (i) parameter encoding by means of a time evolution determined by a quadratic Hamiltonian and parameter encoding into the eigenstates that can be calculated from the measurement basis by using a parameter dependent Bogoliubov transformation (ii) with or (iii) without particle number conservation. The authors find an analytical formula for the QFI that covers the above mentioned cases and apply it to the quantum Hall system (rectangle $w\times L$). In the latter the parameter is the magnetic field (cyclotron frequency) and the state is the ground state (zero temperature) with a particular filling and particle number $N$. The authors show that the QFI exhibits in leading order in $N$ an algebraic scaling that can exceed $N^2$ and is bounded by $N^3$ for different scaling of the length $w$ and $L$ with $N$ (but $wL\propto N$).

I think the main part of the draft is very technical and the writing style is somewhat complicated. Some quantities are introduced and calculated without even mentioning why they are of relevance for the reader. Later on one realizes that those quantities are actually really useful and this is why I would recommend sometimes to mention why one requires certain quantities before actually introducing them. However, this might also be a matter of style and in general I find the topic of this work very interesting and very detailed. In addition, I believe that the results are new and interesting for the broad readership of Sci Post Physics. Before supporting publication in Sci Post Physics I would like the authors to address several points below.

Requested changes

(1) In the introduction in (ii) the authors mention that an infinitesimal amount of relaxation can suffice to relax the system back into its ground state after an infinitesimal parameter change. This is why one can use the instantaneous parameter-dependent ground state. This sounds strange to me since I would in general also expect the relaxation itself to modify the QFI. Why is this not the case? Do you require fast relaxation?

(2) In the introduction: if I understood it correctly the difference between (ii) and (iii) is particle number conservation. You claim that this [...] opens the way to a new kind of quantum parameter estimation not possible with single-particle basis change.'' Could the authors specify what they mean? What is not possible with conserved particle number that is possible without? This seems to be a strong statement but in the end the authors do not need it for the quantum Hall system.

(3) Section II is in general only needed for reading Section IV and as stand-alone not new or motivated. Still I find it useful since it clarifies many technical steps. I am wondering if the authors could switch Sec. II and Sec. III. I have the impression the main goal of this paper is to calculate the QFI which is explained in Sec. III and for this one needs to calculate the overlaps of the Gaussian states which is shown before in Sec. II.

(4) In Sec. II the authors do not define $\alpha^\dagger$ is it the transpose of the vector and conjugate of the elements? It is also sometimes hard to read what is a vector operator or scalar in the paper. Could the authors improve that?

(5) In Sec. III 3. the authors motivate the ''chain rule'' by mentioning it will be natural if one looks at Sec. IV B. I would prefer to get a better motivation maybe by mentioning what was stated in the introduction: one can have different types of parameter encoding by means of time evolution (i) and parameter dependent states (ii), (iii).

(6) The paragraph 3 ends with ''Recently, in [23] a chain rule was derived [...]''. I am wondering what the purpose of this sentence is? Is this chain rule the same?

(7) Before Eq.(83): ''Deriving (74) [...]'' sounds strange to me. Do the authors mean ''Performing the derivative of Eq. (74) [...]''?

(8) In Sec. IV B can the authors mention that to obtain Eq.(88) one uses Eqs. (84) and (86)?

(9) Eq. (98): Is there an imaginary unit i missing? Also, what is the $\bar{n}$ with $c$ in the exponent?

(10) Deriving the summation in Eq. (108): Can the authors add a sentence to clarify the limits of the sums?

(11) Before Eq. (114) can the auhors add ''[...] leading term in $N$ [...]''?

(12) Below Eq. (114). What does Heisenberg scaling mean for this system? Is a scaling of $N^2$ any special? Often for spin systems Heisenberg scaling is the fundamental limit that one can possibly achieve and one needs entanglement for it. Here, the authors claim to beat the Heisenberg limit (without entanglement?). What is the fundamental limit then?

(13) Figure 1 is very hard to look at. I would suggest to make this a color plot with a colorbar. In addition, I would suggest to add more text for the description and discussion of this plot.

---

## Round 1 · Referee Report · Anonymous (Referee 2) · 2023-8-26

Report

The paper analyzes the quantum fisher information (QFI) of a 2DEG, aiming to obtain ultimate theoretical bounds on the measurement precision of magnetic field in Hall sensors. It is well written, providing a comprehensive and self-standing discussion of the problem. In particular, it gives a general and elegant discussion of Bogoliubov transformation and related basis transformations within second quantization, derives formulas for QFI related to such changes, and finally applies those formulas to 2DEG in a magnetic field. The authors claim that the QFI bound on sensitivity follows so-called Heisenberg or even "super-Heisenberg" scaling with the number of particles in the setup. The work opens a pathway for multiple follow-up works.

My main issue with the paper is related to the latter claims of "super-Heisenberg" scaling, which I believe are interpretational misunderstandings. Namely, in the introduction the authors discuss three ways to make the state of the system dependent on parameter-to-be-estimated, where the strong dependence translates to better sensitivity of a quantum system as a probe. (i) One is time evolution generated by parameter-dependent Hamiltonian. ii) and iii) are changes of single-particle basis states or Bogoliubov transformations of fermionic modes (the latter includes the former as a special case), where the state of the system instantaneously follows, say, the many-body---and parameter-to-be-estimated dependent---ground state of the setup (a strong assumption of an unspecified dissipation that relaxes the system to strongly varying family of ground states without further affecting them; the discussion in the paper is limited to pure states).
The 2DEG results in Sec. V follows the (ii/iii) scenario.

I should mention here an extensive line of works following Zanardi et al. PRA 78, 042105 (2008) [see this work and multiple works citing it], that follows a similar general scheme (mostly aiming to enhance sensitivity through a strong parameter dependence of a many-body ground state around some critical point). There are strong conceptual and technical similarities with the current paper -- even for spin models that in some often-studied cases are conveniently studied via mapping to fermionic systems. Some of those works should likely be cited here.

The authors treat (i) and (ii/iii) on the same footing, while there are fundamental differences between those two classes when it comes to how the time appears in the picture. The Heisenberg scaling appears in (i) basically as a theorem, see e.q. Ref.[24] of the current paper, where the Heisenberg scaling is in Eq. (15) with k=0 in (16) (clear assumption, satisfied also in this work). The important part here is that the bound on sensitivity improves linearly in time, with particle-number dependence (and so-called Heisenberg scaling) appearing in the prefactor, i.e. in the rate at which the bound on sensitivity improves with time. On the other hand, in (ii/iii) there is no time. One cannot treat "rate of change" and "change" on the same footing and hence talking about "super-Heisenberg" scaling in (ii/iii) is an abuse of terminology in my opinion.

In fact, (ii/iii) can be treated as a special case of (i) with time-dependent Hamiltonian assuming slow adiabatic dynamics -- which is conceptually clear for pure systems. Then, the time needed for adiabatic changes might scale with the number of constituents (or timescale of other relaxation processes), affecting how we interpret the scaling, see Rams et al, PRX 8, 021022 (2018). This is in clear contradiction with some elements of discussion in the first paragraph on page 2.

I believe the related aspects of the discussion should be cleared.

Besides:
- I don't understand what physical process/scenario the authors cover in Sec. IV B.
The section is entitled "Hamiltonian evolution", but the Hamiltonian evolution would not have \hat T in Eq. (75) with canonical results, e.g., in Ref. [24].
Is it a scenario assuming some kind of instantaneous relaxation to new basis states, followed by free evolution? Can the addition of \hat T in (75) be reasonably justified for generic state |psi(0)>?

On the other hand, the theoretical derivations in Sec IV B, C, and IIIC3 seem redundant from the perspective of the physical example discussed in Sec V.
Unless the authors have some future directions in mind where those can be useful, I would suggest that those sections can well be removed to keep the discussion focused and clear.

-- The values in Figure 1. are unreadable. I would recommend changing the figure to a color-coded 2D plot (with z-axis encoded as color) or a standard log-log plot with a few lines for selected B's.

-- there is a natural question if experimentally relevant observables allow saturating the QFI bound. Can anything be said about it here?

-- there seems to be inconsistency in the square roots between Eq. (30) and the inline equation just above it.

-- specifying the order of creation operators in (2) might be good to fix convention (like signs in equations (63-68))

---

## Round 2 · Referee Report · Anonymous (Referee 1) · 2024-2-6

Report

In the red-marked resubmitted manuscript I have seen most of my previously asked questions answered. However, I would have appreciated if the authors would have written a proper response since this would have facilitated my second review. Nevertheless, I now support publication of this manuscript in SciPost Physics.

---

## Round 2 · Referee Report · Anonymous (Referee 2) · 2024-2-17

Report

The author's response to the referee's comments, quite unusually, is contained in highlighted parts of the modified text. Those modifications, however, satisfactorily address all my previous comments.

The article gives a detailed and comprehensive analysis of the theoretical bounds on the accuracy of 2DEG as a magnetic field sensor. In my view, this fulfills the acceptance criteria for SciPost Physics, and with the last round of modifications the paper is ready for publication.

---

## Round 2 · Author Response

We thank both Referees for their careful assessment of our work. We modified the manuscript to take into account all of the Referee's observations. This has given us the opportunity to be more precise on various aspects our our work and to enrich the paper with results closer to experimental preoccupations.

---

## Round 2 · List of Changes

Changes are highlighted in red in the new version of the manuscript

---

## Editorial Decision

published